# Tackling the Generative Learning Trilemma with Denoising Diffusion GANs

**Zhisheng Xiao**[*]
The University of Chicago
zxiao@uchicago.edu

**Karsten Kreis**
NVIDIA
kkreis@nvidia.com

**Arash Vahdat**
NVIDIA
avahdat@nvidia.com

## Abstract

A wide variety of deep generative models has been developed in the past decade. Yet, these models often struggle with simultaneously addressing three key requirements including: high sample quality, mode coverage, and fast sampling. We call the challenge imposed by these requirements *the generative learning trilemma*, as the existing models often trade some of them for others. Particularly, denoising diffusion models have shown impressive sample quality and diversity, but their expensive sampling does not yet allow them to be applied in many real-world applications. In this paper, we argue that slow sampling in these models is fundamentally attributed to the Gaussian assumption in the denoising step which is justified only for small step sizes. To enable denoising with large steps, and hence, to reduce the total number of denoising steps, we propose to model the denoising distribution using a complex multimodal distribution. We introduce *denoising diffusion generative adversarial networks (denoising diffusion GANs)* that model each denoising step using a multimodal conditional GAN. Through extensive evaluations, we show that denoising diffusion GANs obtain sample quality and diversity competitive with original diffusion models while being $2000\times$ faster on the CIFAR-10 dataset. Compared to traditional GANs, our model exhibits better mode coverage and sample diversity. To the best of our knowledge, denoising diffusion GAN is the first model that reduces sampling cost in diffusion models to an extent that allows them to be applied to real-world applications inexpensively. Project page and code: https://nvlabs.github.io/denoising-diffusion-gan.

## 1 Introduction

In the past decade, a plethora of deep generative models has been developed for various domains such as images (Karras et al., 2019; Razavi et al., 2019), audio (Oord et al., 2016a; Kong et al., 2021), point clouds (Yang et al., 2019) and graphs (De Cao & Kipf, 2018). However, current generative learning frameworks cannot yet simultaneously satisfy three key requirements, often needed for their wide adoption in real-world problems. These requirements include (i) high-quality sampling, (ii) mode coverage and sample diversity, and (iii) fast and computationally inexpensive sampling. For example, most current works in image synthesis focus on high-quality generation. However, mode coverage and data

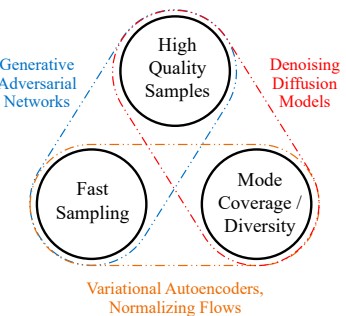

Figure 1: Generative learning trilemma.

diversity are important for better representing minorities and for reducing the negative social impacts of generative models. Additionally, applications such as interactive image editing or real-time speech synthesis require fast sampling. Here, we identify the challenge posed by these requirements as the *generative learning trilemma*, since existing models usually compromise between them.

Fig. 1 summarizes how mainstream generative frameworks tackle the trilemma. Generative adversarial networks (GANs) (Goodfellow et al., 2014; Brock et al., 2018) generate high-quality samples rapidly, but they have poor mode coverage (Salimans et al., 2016; Zhao et al., 2018). Conversely,

---

[*]Work done during an internship at NVIDIA.

variational autoencoders (VAEs) (Kingma & Welling, 2014; Rezende et al., 2014) and normalizing flows (Dinh et al., 2016; Kingma & Dhariwal, 2018) cover data modes faithfully, but they often suffer from low sample quality. Recently, diffusion models (Sohl-Dickstein et al., 2015; Ho et al., 2020; Song et al., 2021c) have emerged as powerful generative models. They demonstrate surprisingly good results in sample quality, beating GANs in image generation (Dhariwal & Nichol, 2021; Ho et al., 2021). They also obtain good mode coverage, indicated by high likelihood (Song et al., 2021b; Kingma et al., 2021; Huang et al., 2021). Although diffusion models have been applied to a variety of tasks (Dhariwal & Nichol; Austin et al.; Mittal et al.; Luo & Hu), sampling from them often requires thousands of network evaluations, making their application expensive in practice.

In this paper, we tackle the generative learning trilemma by reformulating denoising diffusion models specifically for fast sampling while maintaining strong mode coverage and sample quality. We investigate the slow sampling issue of diffusion models and we observe that diffusion models commonly assume that the denoising distribution can be approximated by Gaussian distributions. However, it is known that the Gaussian assumption holds only in the infinitesimal limit of small denoising steps (Sohl-Dickstein et al., 2015; Feller, 1949), which leads to the requirement of a large number of steps in the reverse process. When the reverse process uses larger step sizes (i.e., it has fewer denoising steps), we need a non-Gaussian multimodal distribution for modeling the denoising distribution. Intuitively, in image synthesis, the multimodal distribution arises from the fact that multiple plausible clean images may correspond to the same noisy image.

Inspired by this observation, we propose to parametrize the denoising distribution with an expressive multimodal distribution to enable denoising for large steps. In particular, we introduce a novel generative model, termed as *denoising diffusion GAN*, in which the denoising distributions are modeled with conditional GANs. In image generation, we observe that our model obtains sample quality and mode coverage competitive with diffusion models, while taking only as few as two denoising steps, achieving about $2000\times$ speed-up in sampling compared to the predictor-corrector sampling by Song et al. (2021c) on CIFAR-10. Compared to traditional GANs, we show that our model significantly outperforms state-of-the-art GANs in sample diversity, while being competitive in sample fidelity.

In summary, we make the following contributions: i) We attribute the slow sampling of diffusion models to the Gaussian assumption in the denoising distribution and propose to employ complex, multimodal denoising distributions. ii) We propose denoising diffusion GANs, a diffusion model whose reverse process is parametrized by conditional GANs. iii) Through careful evaluations, we demonstrate that denoising diffusion GANs achieve several orders of magnitude speed-up compared to current diffusion models for both image generation and editing. We show that our model overcomes the deep generative learning trilemma to a large extent, making diffusion models for the first time applicable to interactive, real-world applications at a low computational cost.

## 2 BACKGROUND

In diffusion models (Sohl-Dickstein et al., 2015; Ho et al., 2020), there is a forward process that gradually adds noise to the data $\mathbf{x}_0 \sim q(\mathbf{x}_0)$ in $T$ steps with pre-defined variance schedule $\beta_t$:

$$q(\mathbf{x}_{1:T}|\mathbf{x}_0) = \prod_{t \geq 1} q(\mathbf{x}_t|\mathbf{x}_{t-1}), \quad q(\mathbf{x}_t|\mathbf{x}_{t-1}) = \mathcal{N}(\mathbf{x}_t; \sqrt{1-\beta_t}\mathbf{x}_{t-1}, \beta_t\mathbf{I}), \tag{1}$$

where $q(\mathbf{x}_0)$ is a data-generating distribution. The reverse denoising process is defined by:

$$p_\theta(\mathbf{x}_{0:T}) = p(\mathbf{x}_T)\prod_{t \geq 1} p_\theta(\mathbf{x}_{t-1}|\mathbf{x}_t), \quad p_\theta(\mathbf{x}_{t-1}|\mathbf{x}_t) = \mathcal{N}(\mathbf{x}_{t-1}; \boldsymbol{\mu}_\theta(\mathbf{x}_t, t), \sigma_t^2\mathbf{I}), \tag{2}$$

where $\boldsymbol{\mu}_\theta(\mathbf{x}_t, t)$ and $\sigma_t^2$ are the mean and variance for the denoising model and $\theta$ denotes its parameters. The goal of training is to maximize the likelihood $p_\theta(\mathbf{x}_0) = \int p_\theta(\mathbf{x}_{0:T})d\mathbf{x}_{1:T}$, by maximizing the evidence lower bound (ELBO, $\mathcal{L} \leq \log p_\theta(\mathbf{x}_0)$). The ELBO can be written as matching the true denoising distribution $q(\mathbf{x}_{t-1}|\mathbf{x}_t)$ with the parameterized denoising model $p_\theta(\mathbf{x}_{t-1}|\mathbf{x}_t)$ using:

$$\mathcal{L} = -\sum_{t \geq 1} \mathbb{E}_{q(\mathbf{x}_t)}\left[D_{\mathrm{KL}}\left(q(\mathbf{x}_{t-1}|\mathbf{x}_t)\|p_\theta(\mathbf{x}_{t-1}|\mathbf{x}_t)\right)\right] + C, \tag{3}$$

where $C$ contains constant terms that are independent of $\theta$ and $D_{\mathrm{KL}}$ denotes the Kullback-Leibler (KL) divergence. The objective above is intractable due to the unavailability of $q(\mathbf{x}_{t-1}|\mathbf{x}_t)$. Instead,

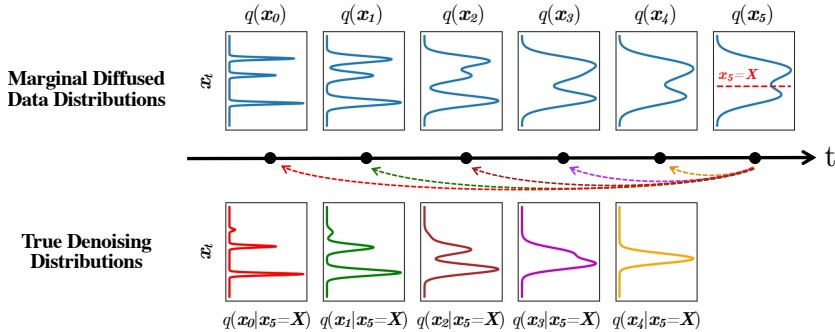

Figure 2: **Top**: The evolution of 1D data distribution $q(\mathbf{x}_0)$ through the diffusion process. **Bottom**:, The visualization of the true denoising distribution for varying step sizes conditioned on a fixed $\mathbf{x}_5$. The true denoising distribution for a small step size (i.e., $q(\mathbf{x}_4|\mathbf{x}_5 = X)$) is close to a Gaussian distribution. However, it becomes more complex and multimodal as the step size increases.

Sohl-Dickstein et al. (2015) show that $\mathcal{L}$ can be written in an alternative form with tractable distributions (see Appendix A of Ho et al. (2020) for details). Ho et al. (2020) show the equivalence of this form with score-based models trained with denoising score matching (Song & Ermon, 2019; 2020).

Two key assumptions are commonly made in diffusion models: First, the denoising distribution $p_\theta(\mathbf{x}_{t-1}|\mathbf{x}_t)$ is modeled with a Gaussian distribution. Second, the number of denoising steps $T$ is often assumed to be in the order of hundreds to thousands of steps. In this paper, we focus on discrete-time diffusion models. In continuous-time diffusion models (Song et al., 2021c), similar assumptions are also made at the sampling time when discretizing time into small timesteps.

## 3 DENOISING DIFFUSION GANS

We first discuss why reducing the number of denoising steps requires learning a multimodal denoising distribution in Sec. 3.1. Then, we present our multimodal denoising model in Sec. 3.2.

### 3.1 MULTIMODAL DENOISING DISTRIBUTIONS FOR LARGE DENOISING STEPS

As we discussed in Sec. 2, a common assumption in the diffusion model literature is to approximate $q(\mathbf{x}_{t-1}|\mathbf{x}_t)$ with a Gaussian distribution. Here, we question when such an approximation is accurate.

The true denoising distribution $q(\mathbf{x}_{t-1}|\mathbf{x}_t)$ can be written as $q(\mathbf{x}_{t-1}|\mathbf{x}_t) \propto q(\mathbf{x}_t|\mathbf{x}_{t-1})q(\mathbf{x}_{t-1})$ using Bayes' rule where $q(\mathbf{x}_t|\mathbf{x}_{t-1})$ is the forward Gaussian diffusion shown in Eq. 1 and $q(\mathbf{x}_{t-1})$ is the marginal data distribution at step $t$. It can be shown that in two situations the true denoising distribution takes a Gaussian form. First, in the limit of infinitesimal step size $\beta_t$, the product in the Bayes' rule is dominated by $q(\mathbf{x}_t|\mathbf{x}_{t-1})$ and the reversal of the diffusion process takes an identical functional form as the forward process (Feller, 1949). Thus, when $\beta_t$ is sufficiently small, since $q(\mathbf{x}_t|\mathbf{x}_{t-1})$ is a Gaussian, the denoising distribution $q(\mathbf{x}_{t-1}|\mathbf{x}_t)$ is also Gaussian, and the approximation used by current diffusion models can be accurate. To satisfy this, diffusion models often have thousands of steps with small $\beta_t$. Second, if data marginal $q(\mathbf{x}_t)$ is Gaussian, the denoising distribution $q(\mathbf{x}_{t-1}|\mathbf{x}_t)$ is also a Gaussian distribution. The idea of bringing data distribution $q(\mathbf{x}_0)$ and consequently $q(\mathbf{x}_t)$ closer to Gaussian using a VAE encoder was recently explored in LSGM (Vahdat et al., 2021). However, the problem of transforming the data to Gaussian itself is challenging and VAE encoders cannot solve it perfectly. That is why LSGM still requires tens to hundreds of steps on complex datasets.

In this paper, we argue that when neither of the conditions are met, i.e., when the denoising step is large and the data distribution is non-Gaussian, there are no guarantees that the Gaussian assumption on the denoising distribution holds. To illustrate this, in Fig. 2, we visualize the true denoising distribution for different denoising step sizes for a multimodal data distribution. We see that as the denoising step gets larger, the true denoising distribution becomes more complex and multimodal.

### 3.2 MODELING DENOISING DISTRIBUTIONS WITH CONITIONAL GANS

Our goal is to reduce the number of denoising diffusion steps $T$ required in the reverse process of diffusion models. Inspired by the observation above, we propose to model the denoising distribution

with an expressive multimodal distribution. Since conditional GANs have been shown to model complex conditional distributions in the image domain (Mirza & Osindero, 2014; Ledig et al., 2017; Isola et al., 2017), we adopt them to approximate the true denoising distribution $q(\mathbf{x}_{t-1}|\mathbf{x}_t)$.

Specifically, our forward diffusion is set up similarly to the diffusion models in Eq. 1 with the main assumption that $T$ is assumed to be small ($T \leq 8$) and each diffusion step has larger $\beta_t$. Our training is formulated by matching the conditional GAN generator $p_\theta(\mathbf{x}_{t-1}|\mathbf{x}_t)$ and $q(\mathbf{x}_{t-1}|\mathbf{x}_t)$ using an adversarial loss that minimizes a divergence $D_{\text{adv}}$ per denoising step:

$$\min_\theta \sum_{t \geq 1} \mathbb{E}_{q(\mathbf{x}_t)} \left[ D_{\text{adv}}(q(\mathbf{x}_{t-1}|\mathbf{x}_t) \| p_\theta(\mathbf{x}_{t-1}|\mathbf{x}_t)) \right], \tag{4}$$

where $D_{\text{adv}}$ can be Wasserstein distance, Jenson-Shannon divergence, or f-divergence depending on the adversarial training setup (Arjovsky et al., 2017; Goodfellow et al., 2014; Nowozin et al., 2016). In this paper, we rely on non-saturating GANs (Goodfellow et al., 2014), that are widely used in successful GAN frameworks such as StyleGANs (Karras et al., 2019; 2020b). In this case, $D_{\text{adv}}$ takes a special instance of f-divergence called *softened reverse KL* (Shannon et al., 2020), which is different from the forward KL divergence used in the original diffusion model training in Eq. 3[1].

To set up the adversarial training, we denote the time-dependent discriminator as $D_\phi(\mathbf{x}_{t-1}, \mathbf{x}_t, t)$ : $\mathbb{R}^N \times \mathbb{R}^N \times \mathbb{R} \to [0, 1]$, with parameters $\phi$. It takes the $N$-dimensional $\mathbf{x}_{t-1}$ and $\mathbf{x}_t$ as inputs, and decides whether $\mathbf{x}_{t-1}$ is a plausible denoised version of $\mathbf{x}_t$. The discriminator is trained by:

$$\min_\phi \sum_{t \geq 1} \mathbb{E}_{q(\mathbf{x}_t)} \left[ \mathbb{E}_{q(\mathbf{x}_{t-1}|\mathbf{x}_t)} [-\log(D_\phi(\mathbf{x}_{t-1}, \mathbf{x}_t, t)] + \mathbb{E}_{p_\theta(\mathbf{x}_{t-1}|\mathbf{x}_t)} [-\log(1 - D_\phi(\mathbf{x}_{t-1}, \mathbf{x}_t, t))] \right], \tag{5}$$

where fake samples from $p_\theta(\mathbf{x}_{t-1}|\mathbf{x}_t)$ are contrasted against real samples from $q(\mathbf{x}_{t-1}|\mathbf{x}_t)$. The first expectation requires sampling from $q(\mathbf{x}_{t-1}|\mathbf{x}_t)$ which is unknown. However, we use the identity $q(\mathbf{x}_t, \mathbf{x}_{t-1}) = \int d\mathbf{x}_0 q(\mathbf{x}_0) q(\mathbf{x}_t, \mathbf{x}_{t-1}|\mathbf{x}_0) = \int d\mathbf{x}_0 q(\mathbf{x}_0) q(\mathbf{x}_{t-1}|\mathbf{x}_0) q(\mathbf{x}_t|\mathbf{x}_{t-1})$ to rewrite the first expectation in Eq. 5 as:

$$\mathbb{E}_{q(\mathbf{x}_t)q(\mathbf{x}_{t-1}|\mathbf{x}_t)} [-\log(D_\phi(\mathbf{x}_{t-1}, \mathbf{x}_t, t))] = \mathbb{E}_{q(\mathbf{x}_0)q(\mathbf{x}_{t-1}|\mathbf{x}_0)q(\mathbf{x}_t|\mathbf{x}_{t-1})} [-\log(D_\phi(\mathbf{x}_{t-1}, \mathbf{x}_t, t))].$$

Given the discriminator, we train the generator by $\max_\theta \sum_{t \geq 1} \mathbb{E}_{q(\mathbf{x}_t)} \mathbb{E}_{p_\theta(\mathbf{x}_{t-1}|\mathbf{x}_t)} [\log(D_\phi(\mathbf{x}_{t-1}, \mathbf{x}_t, t))]$, which updates the generator with the non-saturating GAN objective (Goodfellow et al., 2014).

**Parametrizing the implicit denoising model:** Instead of directly predicting $\mathbf{x}_{t-1}$ in the denoising step, diffusion models (Ho et al., 2020) can be interpreted as parameterizing the denoising model by $p_\theta(\mathbf{x}_{t-1}|\mathbf{x}_t) := q(\mathbf{x}_{t-1}|\mathbf{x}_t, \mathbf{x}_0 = f_\theta(\mathbf{x}_t, t))$ in which first $\mathbf{x}_0$ is predicted using the denoising model $f_\theta(\mathbf{x}_t, t)$, and then, $\mathbf{x}_{t-1}$ is sampled using the posterior distribution $q(\mathbf{x}_{t-1}|\mathbf{x}_t, \mathbf{x}_0)$ given $\mathbf{x}_t$ and the predicted $\mathbf{x}_0$ (See Appendix B for details). The distribution $q(\mathbf{x}_{t-1}|\mathbf{x}_0, \mathbf{x}_t)$ is intuitively the distribution over $\mathbf{x}_{t-1}$ when denoising from $\mathbf{x}_t$ towards $\mathbf{x}_0$, and it always has a Gaussian form for the diffusion process in Eq. 1, independent of the step size and complexity of the data distribution (see Appendix A for the expression of $q(\mathbf{x}_{t-1}|\mathbf{x}_0, \mathbf{x}_t)$). Similarly, we define $p_\theta(\mathbf{x}_{t-1}|\mathbf{x}_t)$ by:

$$p_\theta(\mathbf{x}_{t-1}|\mathbf{x}_t) := \int p_\theta(\mathbf{x}_0|\mathbf{x}_t) q(\mathbf{x}_{t-1}|\mathbf{x}_t, \mathbf{x}_0) d\mathbf{x}_0 = \int p(\mathbf{z}) q(\mathbf{x}_{t-1}|\mathbf{x}_t, \mathbf{x}_0 = G_\theta(\mathbf{x}_t, \mathbf{z}, t)) d\mathbf{z}, \tag{6}$$

where $p_\theta(\mathbf{x}_0|\mathbf{x}_t)$ is the implicit distribution imposed by the GAN generator $G_\theta(\mathbf{x}_t, \mathbf{z}, t) : \mathbb{R}^N \times \mathbb{R}^L \times \mathbb{R} \to \mathbb{R}^N$ that outputs $\mathbf{x}_0$ given $\mathbf{x}_t$ and an $L$-dimensional latent variable $\mathbf{z} \sim p(\mathbf{z}) := \mathcal{N}(\mathbf{z}; \mathbf{0}, \mathbf{I})$.

Our parameterization has several advantages: Firstly, our $p_\theta(\mathbf{x}_{t-1}|\mathbf{x}_t)$ is formulated similar to DDPM (Ho et al., 2020). Thus, we can borrow some inductive bias such as the network structure design from DDPM. The main difference is that, in DDPM, $\mathbf{x}_0$ is predicted as a deterministic mapping of $\mathbf{x}_t$, while in our case $\mathbf{x}_0$ is produced by the generator with random latent variable $\mathbf{z}$. This is the key difference that allows our denoising distribution $p_\theta(\mathbf{x}_{t-1}|\mathbf{x}_t)$ to become multimodal and complex in contrast to the unimodal denoising model in DDPM. Secondly, note that for different $t$'s, $\mathbf{x}_t$ has different levels of perturbation, and hence using a single network to predict $\mathbf{x}_{t-1}$ directly at different $t$ may be difficult. However, in our case the generator only needs to predict unperturbed $\mathbf{x}_0$ and then add back perturbation using $q(\mathbf{x}_{t-1}|\mathbf{x}_t, \mathbf{x}_0)$. Fig. 3 visualizes our training pipeline.

---

[1]At early stages, we examined training a conditional VAE as $p_\theta(\mathbf{x}_{t-1}|\mathbf{x}_t)$ by minimizing $\sum_{t=1}^{T} \mathbb{E}_{q(t)} [D_{\text{KL}}(q(\mathbf{x}_{t-1}|\mathbf{x}_t) \| p_\theta(\mathbf{x}_{t-1}|\mathbf{x}_t))]$ for small $T$. However, conditional VAEs consistently resulted in poor generative performance in our early experiments. In this paper, we focus on conditional GANs and we leave the exploration of other expressive conditional generators for $p_\theta(\mathbf{x}_{t-1}|\mathbf{x}_t)$ to future work.

**Advantage over one-shot generator:** One natural question for our model is, why not just train a GAN that can generate samples in one shot using a traditional setup, in contrast to our model that generates samples by denoising iteratively? Our model has several advantages over traditional GANs. GANs are known to suffer from training instability and mode collapse (Kodali et al., 2017; Salimans et al., 2016), and some possible reasons include the difficulty of directly generating samples from a complex distribution in one-shot, and the overfitting issue when the discriminator only looks at clean samples. In contrast, our model breaks the generation process into several conditional denoising diffusion steps in which each step is relatively simple to model, due to the strong conditioning on $\mathbf{x}_t$. Moreover, the diffusion process smoothens the data distribution (Lyu, 2012), making the discriminator less likely to overfit. Thus, we expect our model to exhibit better training stability and mode coverage. We empirically verify the advantages over traditional GANs in Sec. 5.

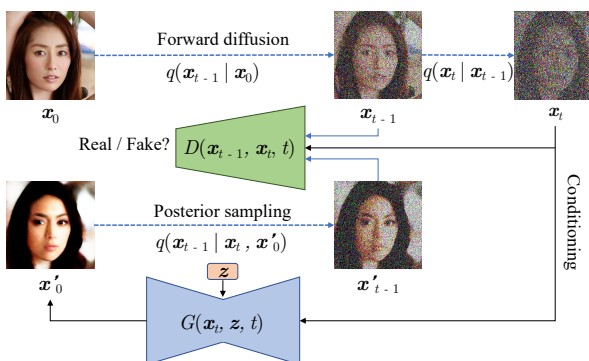

Figure 3: The training process of denoising diffusion GAN.

## 4 RELATED WORK

Diffusion-based models (Sohl-Dickstein et al., 2015; Ho et al., 2020) learn the finite-time reversal of a diffusion process, sharing the idea of learning transition operators of Markov chains with Goyal et al. (2017); Alain et al. (2016); Bordes et al. (2017). Since then, there have been a number of improvements and alternatives to diffusion models. Song et al. (2021c) generalize diffusion processes to continuous time, and provide a unified view of diffusion models and denoising score matching (Vincent, 2011; Song & Ermon, 2019). Jolicoeur-Martineau et al. (2021b) add an auxiliary adversarial loss to the main objective. This is fundamentally different from ours, as their auxiliary adversarial loss only acts as an image enhancer, and they do not use latent variables; therefore, the denoising distribution is still a unimodal Gaussian. Other explorations include introducing alternative noise distributions in the forward process (Nachmani et al., 2021), jointly optimizing the model and noise schedule (Kingma et al., 2021) and applying the model in latent spaces (Vahdat et al., 2021).

One major drawback of diffusion or score-based models is the slow sampling speed due to a large number of iterative sampling steps. To alleviate this issue, multiple methods have been proposed, including knowledge distillation (Luhman & Luhman, 2021), learning an adaptive noise schedule (San-Roman et al., 2021), introducing non-Markovian diffusion processes (Song et al., 2021a; Kong & Ping, 2021), and using better SDE solvers for continuous-time models (Jolicoeur-Martineau et al., 2021a). In particular, Song et al. (2021a) uses $\mathbf{x}_0$ sampling as a crucial ingredient to their method, but their denoising distribution is still a Gaussian. These methods either suffer from significant degradation in sample quality, or still require many sampling steps as we demonstrate in Sec. 5.

Among variants of diffusion models, Gao et al. (2021) have the closest connection with our method. They propose to model the single-step denoising distribution by a conditional energy-based model (EBM), sharing the high-level idea of using expressive denoising distributions with us. However, they motivate their method from the perspective of facilitating the training of EBMs. More importantly, although only a few denoising steps are needed, expensive MCMC has to be used to sample from each denoising step, making the sampling process slow with ∼180 network evaluations. ImageBART (Esser et al., 2021a) explores modeling the denoising distribution of a diffusion process on discrete latent space with an auto-regressive model per step in a few denoising steps. However, the auto-regressive structure of their denoising distribution still makes sampling slow.

Since our model is trained with adversarial loss, our work is related to recent advances in improving the sample quality and diversity of GANs, including data augmentation (Zhao et al., 2020; Karras et al., 2020a), consistency regularization (Zhang et al., 2020; Zhao et al., 2021) and entropy regularization (Dieng et al., 2019). In addition, the idea of training generative models with smoothed distributions is also discussed in Meng et al. (2021a) for auto-regressive models.

# 5 EXPERIMENTS

In this section, we evaluate our proposed denoising diffusion GAN for the image synthesis problem. We begin with briefly introducing the network architecture design, while additional implementation details are presented in Appendix C. For our GAN generator, we adopt the NCSN++ architecture from Song et al. (2021c) which has a U-net structure (Ronneberger et al., 2015). The conditioning $\mathbf{x}_t$ is the input of the network, and time embedding is used to ensure conditioning on $t$. We let the latent variable $\mathbf{z}$ control the normalization layers. In particular, we replace all group normalization layers (Wu & He, 2018) in NCSN++ with adaptive group normalization layers in the generator, similar to Karras et al. (2019); Huang & Belongie (2017), where the shift and scale parameters in group normalization are predicted from $\mathbf{z}$ using a simple multi-layer fully-connected network.

## 5.1 OVERCOMING THE GENERATIVE LEARNING TRILEMMA

One major highlight of our model is that it excels at all three criteria in the generative learning trilemma. Here, we carefully evaluate our model's performances on sample fidelity, sample diversity and sampling time, and benchmark against a comprehensive list of models on the CIFAR-10 dataset.

**Evaluation criteria:** We adopt the commonly used Fréchet inception distance (FID) (Heusel et al., 2017) and Inception Score (IS) (Salimans et al., 2016) for evaluating sample fidelity. We use the training set as a reference to compute the FID, following common practice in the literature (see Ho et al. (2020); Karras et al. (2019) as an example). For sample diversity, we use the improved recall score from Kynkäänniemi et al. (2019), which is an improved version of the original precision and recall metric proposed by Sajjadi et al. (2018). It is shown that an improved recall score reflects how the variation in the generated samples matches that in the training set (Kynkäänniemi et al., 2019). For sampling time, we use the number of function evaluations (NFE) and the clock time when generating a batch of 100 images on a V100 GPU.

**Results:** We present our quantitative results in Table 1. We observe that our sample quality is competitive among the best diffusion models and GANs. Although some variants of diffusion models obtain better IS and FID, they require a large number of function evaluations to generate samples (while we use only 4 denoising steps). For example, our sampling time is about 2000× faster than the predictor-corrector sampling by Song et al. (2021c) and ∼20× faster than FastDDPM (Kong & Ping, 2021). Note that diffusion models can produce samples in fewer steps while trading off the sample quality. To better benchmark our method against existing diffusion models, we plot the FID score versus sampling time of diffusion models by varying the number of denoising steps (or the error tolerance for continuous-time models) in Figure 4. The figure clearly shows the advantage of our model compared to previous diffusion models. When comparing our model to GANs, we observe that only StyleGAN2 with adaptive data augmentation has slightly better sample quality than ours. However, from Table 1, we see that GANs have limited sample diversity, as their recall scores are below 0.5. In contrast, our model obtains a significantly better recall score, even higher than several advanced likelihood-based models, and competitive among diffusion models. We show qualitative samples of CIFAR-10 in Figure 5. In summary, our model simultaneously excels at sample quality, sample diversity, and sampling speed and tackles the generative learning trilemma by a large extent.

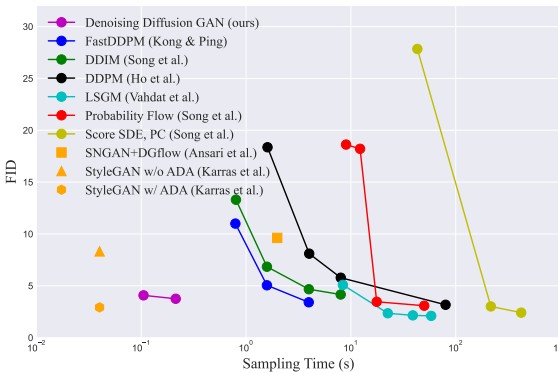
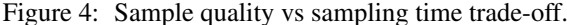

Figure 4: Sample quality vs sampling time trade-off.

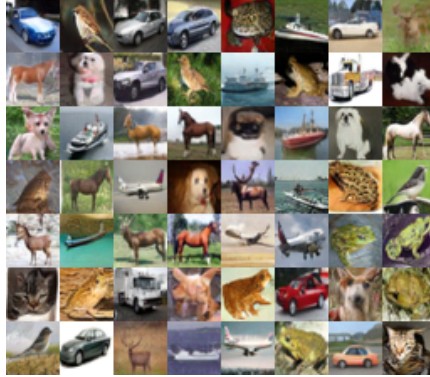

Figure 5: CIFAR-10 qualitative samples.

Table 1: Results for unconditional generation on CIFAR-10.

| Model | IS↑ | FID↓ | Recall↑ | NFE↓ | Time (s)↓ |
|---|---|---|---|---|---|
| Denoising Diffusion GAN (ours), T=4 | 9.63 | 3.75 | 0.57 | 4 | 0.21 |
| DDPM (Ho et al., 2020) | 9.46 | 3.21 | 0.57 | 1000 | 80.5 |
| NCSN (Song & Ermon, 2019) | 8.87 | 25.3 | - | 1000 | 107.9 |
| Adversarial DSM (Jolicoeur-Martineau et al., 2021b) | - | 6.10 | - | 1000 | - |
| Likelihood SDE (Song et al., 2021b) | - | 2.87 | - | - | - |
| Score SDE (VE) (Song et al., 2021c) | 9.89 | 2.20 | 0.59 | 2000 | 423.2 |
| Score SDE (VP) (Song et al., 2021c) | 9.68 | 2.41 | 0.59 | 2000 | 421.5 |
| Probability Flow (VP) (Song et al., 2021c) | 9.83 | 3.08 | 0.57 | 140 | 50.9 |
| LSGM (Vahdat et al., 2021) | 9.87 | 2.10 | 0.61 | 147 | 44.5 |
| DDIM, T=50 (Song et al., 2021a) | 8.78 | 4.67 | 0.53 | 50 | 4.01 |
| FastDDPM, T=50 (Kong & Ping, 2021) | 8.98 | 3.41 | 0.56 | 50 | 4.01 |
| Recovery EBM (Gao et al., 2021) | 8.30 | 9.58 | - | 180 | - |
| Improved DDPM (Nichol & Dhariwal, 2021) | - | 2.90 | - | 4000 | - |
| VDM (Kingma et al., 2021) | - | 4.00 | - | 1000 | - |
| UDM (Kim et al., 2021) | 10.1 | 2.33 | - | 2000 | - |
| D3PMs (Austin et al., 2021) | 8.56 | 7.34 | - | 1000 | - |
| Gotta Go Fast (Jolicoeur-Martineau et al., 2021a) | - | 2.44 | - | 180 | - |
| DDPM Distillation (Luhman & Luhman, 2021) | 8.36 | 9.36 | 0.51 | 1 | - |
| SNGAN (Miyato et al., 2018) | 8.22 | 21.7 | 0.44 | 1 | - |
| SNGAN+DGflow (Ansari et al., 2021) | 9.35 | 9.62 | 0.48 | 25 | 1.98 |
| AutoGAN (Gong et al., 2019) | 8.60 | 12.4 | 0.46 | 1 | - |
| TransGAN (Jiang et al., 2021) | 9.02 | 9.26 | - | 1 | - |
| StyleGAN2 w/o ADA (Karras et al., 2020a) | 9.18 | 8.32 | 0.41 | 1 | 0.04 |
| StyleGAN2 w/ ADA (Karras et al., 2020a) | 9.83 | 2.92 | 0.49 | 1 | 0.04 |
| StyleGAN2 w/ Diffaug (Zhao et al., 2020) | 9.40 | 5.79 | 0.42 | 1 | 0.04 |
| Glow (Kingma & Dhariwal, 2018) | 3.92 | 48.9 | - | 1 | - |
| PixelCNN (Oord et al., 2016b) | 4.60 | 65.9 | - | 1024 | - |
| NVAE (Vahdat & Kautz, 2020) | 7.18 | 23.5 | 0.51 | 1 | 0.36 |
| IGEBM (Du & Mordatch, 2019) | 6.02 | 40.6 | - | 60 | - |
| VAEBM (Xiao et al., 2021) | 8.43 | 12.2 | 0.53 | 16 | 8.79 |

## 5.2 ABLATION STUDIES

Here, we provide additional insights into our model by performing ablation studies.

**Number of denoising steps:** In the first part of Table 2, we study the effect of using a different number of denoising steps ($T$). Note that $T = 1$ corresponds to training an unconditional GAN, as the conditioning $\mathbf{x}_t$ contains almost no information about $\mathbf{x}_0$. We observe that $T = 1$ leads to significantly worse results with low sample diversity, indicated by the low recall score. This confirms the benefits of breaking generation into several denoising steps, especially for improving the sample diversity. When varying $T > 1$, we observe that $T = 4$ gives the best results, whereas there is a slight degrade in performance for larger $T$. We hypothesize that we may require a significantly higher capacity to accommodate larger $T$, as we need a conditional GAN for each denoising step.

**Diffusion as data augmentation:** Our model shares some similarities with recent work on applying data augmentation to GANs (Karras et al., 2020a; Zhao et al., 2020). To study the effect of perturbing inputs, we train a one-shot GAN with our network structure following the protocol in (Zhao et al., 2020) with the forward diffusion process as data augmentation. The result, presented in the second group of Table 2, is significantly worse than our model, indicating that our model is not equivalent to augmenting data before applying the discriminator.

**Parametrization for $p_\theta(\mathbf{x}_{t-1}|\mathbf{x}_t)$:** We study two alternative ways to parametrize the denoising distribution for the same $T = 4$ setting. Instead of letting the generator produce estimated samples of $\mathbf{x}_0$, we set the generator to directly output denoised samples $\mathbf{x}_{t-1}$ without posterior sampling (*direct denoising*), or output the noise $\epsilon_t$ that perturbs a clean image to produce $\mathbf{x}_t$ (*noise generation*). Note that the latter case is closely related to most diffusion models where the network deterministically predicts the perturbation noise. In Table 2, we show that although these alternative parametrizations work reasonably well, our main parametrization outperforms them by a large margin.

**Importance of latent variable:** Removing latent variables $\mathbf{z}$ converts our denoising model to a unimodal distribution. In the last line of Table 2, we study our model's performance without any latent variables $\mathbf{z}$. We see that the sample quality is significantly worse, suggesting the importance of multimodal denoising distributions. In Figure 8, we visualize the effect of latent variables by showing samples of $p_\theta(\mathbf{x}_0|\mathbf{x}_1)$, where $\mathbf{x}_1$ is a fixed noisy observation. We see that while the majority of information in the conditioning $\mathbf{x}_1$ is preserved, the samples are diverse due to the latent variables.

Table 2: Ablation studies on CIFAR-10.

| Model Variants | IS↑ | FID↓ | Recall↑ |
|---|---|---|---|
| T = 1 | 8.93 | 14.6 | 0.19 |
| T = 2 | **9.80** | 4.08 | 0.54 |
| T = 4 | 9.63 | **3.75** | **0.57** |
| T = 8 | 9.43 | 4.36 | 0.56 |
| One-shot w/ aug | 8.96 | 13.2 | 0.25 |
| Direct denoising | 9.10 | 6.03 | 0.53 |
| Noise generation | 8.79 | 8.04 | 0.52 |
| No latent variable | 8.37 | 20.6 | 0.42 |

Table 3: Mode coverage on StackedMNIST.

| Model | Modes↑ | KL↓ |
|---|---|---|
| VEEGAN (Srivastava et al.) | 762 | 2.173 |
| PacGAN (Lin et al.) | 992 | 0.277 |
| PresGAN (Dieng et al.) | **1000** | 0.115 |
| InclusiveGAN (Yu et al.) | 997 | 0.200 |
| StyleGAN2 (Karras et al.) | 940 | 0.424 |
| Adv. DSM (Jolicoeur-Martineau et al.) | **1000** | 1.49 |
| VAEBM (Xiao et al.) | **1000** | 0.087 |
| Denoising Diffusion GAN (ours) | **1000** | **0.071** |

## 5.3 ADDITIONAL STUDIES

**Mode Coverage:** Besides the recall score in Table 1, we also evaluate the mode coverage of our model on the popular 25-Gaussians and StackedMNIST. The 25-Gaussians dataset is a 2-D toy dataset, generated by a mixture of 25 two-dimensional Gaussian distributions, arranged in a grid. We train our denoising diffusion GAN with 4 denoising steps and compare it to other models in Figure 6. We observe that the vanilla GAN suffers severely from mode collapse, and while techniques like WGAN-GP (Gulrajani et al., 2017) improve mode coverage, the sample quality is still limited. In contrast, our model covers all the modes while maintaining high sample quality. We also train a diffusion model and plot the samples generated by 100 and 500 denoising steps. We see that diffusion models require a large number of steps to maintain high sample quality.

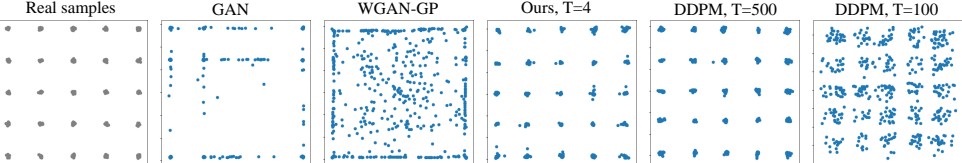

Figure 6: Qualitative results on the 25-Gaussians dataset.

StackMNIST contains images generated by randomly choosing 3 MNIST images and stacking them along the RGB channels. Hence, the data distribution has 1000 modes. Following the setting of Lin et al. (2018), we report the number of covered modes and the KL divergence from the categorical distribution over 1000 categories of generated samples to true data in Table 3. We observe that our model covers all modes faithfully and achieves the lowest KL compared to GANs that are specifically designed for better mode coverage or StyleGAN2 that is known to have the best sample quality.

**Training Stability:** We discuss the training stability of our model in Appendix D.

**High Resolution Images:** We train our model on datasets with larger images, including CelebA-HQ (Karras et al., 2018) and LSUN Church (Yu et al., 2015) at 256 × 256px resolution. We report FID on these two datasets in Table 4 and 5. Similar to CIFAR-10, our model obtains competitive sample quality among the best diffusion models and GANs. In particular, in LSUN Church, our model outperforms DDPM and ImageBART (see Figure 7 and Appendix E for samples). Although, some GANs perform better on this dataset, their mode coverage is not reflected by the FID score.

**Stroke-based image synthesis:** Recently, Meng et al. (2021b) propose an interesting application of diffusion models to stroke-based generation. Specifically, they perturb a stroke painting by the forward diffusion process, and denoise it with a diffusion model. The method is particularly promising because it only requires training an unconditional generative model on the target dataset and does not require training images paired with stroke paintings like GAN-based methods (Sangkloy et al., 2017; Park et al., 2019). We apply our model to stroke-based image synthesis and show qualitative results in Figure 9. The generated samples are realistic and diverse, while the conditioning in the stroke paintings is faithfully preserved. Compared to Meng et al. (2021b), our model enjoys a 1100× speedup in generation, as it takes only **0.16**s to generate one image at 256 resolution vs. **181**s for Meng et al. (2021b). This experiment confirms that our proposed model enables the application of diffusion models to interactive applications such as image editing.

Table 4: Generative results on CelebA-HQ-256

| Model | FID↓ |
|---|---|
| Denoising Diffusion GAN (ours) | 7.64 |
| Score SDE (Song et al., 2021c) | 7.23 |
| LSGM (Vahdat et al., 2021) | 7.22 |
| UDM (Kim et al., 2021) | **7.16** |
| NVAE (Vahdat & Kautz, 2020) | 29.7 |
| VAEBM (Xiao et al., 2021) | 20.4 |
| NCP-VAE (Aneja et al., 2021) | 24.8 |
| PGGAN (Karras et al., 2018) | 8.03 |
| Adv. LAE (Pidhorskyi et al., 2020) | 19.2 |
| VQ-GAN (Esser et al., 2021b) | 10.2 |
| DC-AE (Parmar et al., 2021) | 15.8 |

Table 5: Generative results on LSUN Church 256

| Model | FID↓ |
|---|---|
| Denoising Diffusion GAN (ours) | 5.25 |
| DDPM (Ho et al., 2020) | 7.89 |
| ImageBART (Esser et al., 2021a) | 7.32 |
| Gotta Go Fast (Jolicoeur-Martineau et al.) | 25.67 |
| PGGAN (Karras et al., 2018)) | 6.42 |
| StyleGAN (Karras et al., 2019) | 4.21 |
| StyleGAN2 (Karras et al., 2020b) | 3.86 |
| CIPS (Anokhin et al., 2021) | **2.92** |

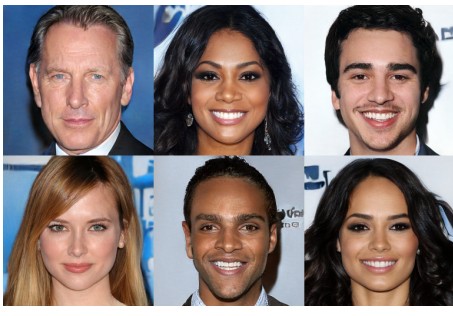 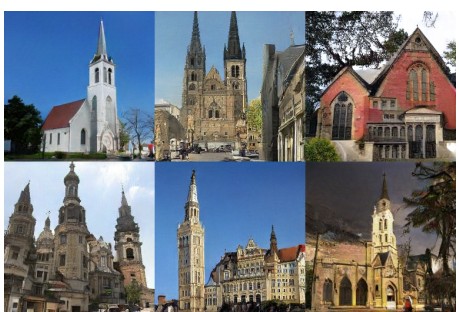

Figure 7: Qualitative results on CelebA-HQ 256 and LSUN Church Outdoor 256.

**Additional results:** Additional qualitative visualizations are provided in Appendix E, F, and G.

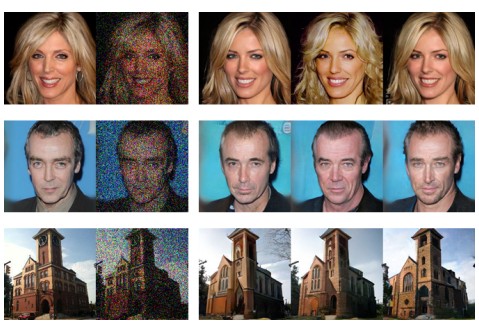 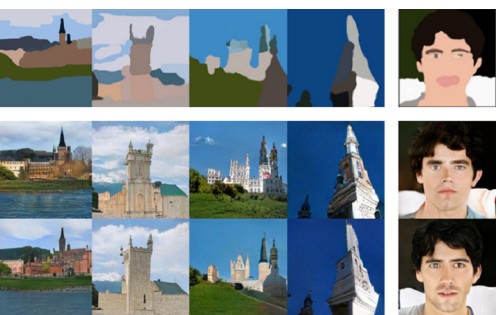

Figure 8: Multi-modality of denoising distribution given the same noisy observation. **Left:** clean image $\mathbf{x}_0$ and perturbed image $\mathbf{x}_1$. **Right:** Three samples from $p_\theta(\mathbf{x}_0|\mathbf{x}_1)$.

Figure 9: Qualitative results on stroke-based synthesis. **Top row:** stroke paintings. **Bottom two rows:** generated samples corresponding to the stroke painting (best seen when zoomed in).

## 6 CONCLUSIONS

Deep generative learning frameworks still struggle with addressing the generative learning trilemma. Diffusion models achieve particularly high-quality and diverse sampling. However, their slow sampling and high computational cost do not yet allow them to be widely applied in real-world applications. In this paper, we argued that one of the main sources of slow sampling in diffusion models is the Gaussian assumption in the denoising distribution, which is justified only for very small denoising steps. To remedy this, we proposed denoising diffusion GANs that model each denoising step using a complex multimodal distribution, enabling us to take large denoising steps. In extensive experiments, we showed that denoising diffusion GANs achieve high sample quality and diversity competitive to the original diffusion models, while being orders of magnitude faster at sampling. Compared to traditional GANs, our proposed model enjoys better mode coverage and sample diversity. Our denoising diffusion GAN overcomes the generative learning trilemma to a large extent, allowing diffusion models to be applied to real-world problems with low computational cost.

## 7    ETHICS AND REPRODUCIBILITY STATEMENT

Generating high-quality samples while representing the diversity in the training data faithfully has been a daunting challenge in generative learning. Mode coverage and high diversity are key requirements for reducing biases in generative models and improving the representation of minorities in a population. While diffusion models achieve both high sample quality and diversity, their expensive sampling limits their application in many real-world problems. Our proposed denoising diffusion GAN reduces the computational complexity of diffusion models to an extent that allows these models to be applied in practical applications at a low cost. Thus, we foresee that in the long term, our model can help with reducing the negative social impacts of existing generative models that fall short in capturing the data diversity.

We evaluate our model using public datasets, and therefore this work does not involve any human subject evaluation or new data collection.

**Reproducibility Statement:**   We currently provide experimental details in the appendices with a detailed list of hyperparameters and training settings. To aid with reproducibility further, we will release our source code publicly in the future with instructions to reproduce our results.

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

## A    DERIVATION FOR THE GAUSSIAN POSTERIOR

Ho et al. (2020) provide a derivation for the Gaussian posterior distribution. We include it here for completeness. Consider the forward diffusion process in Eq. 1, which we repeat here:

$$q(\mathbf{x}_{1:T}|\mathbf{x}_0) = \prod_{t=1}^{T} q(\mathbf{x}_t|\mathbf{x}_{t-1}), \quad q(\mathbf{x}_t|\mathbf{x}_{t-1}) = \mathcal{N}(\mathbf{x}_t; \sqrt{1-\beta_t}\mathbf{x}_{t-1}, \beta_t \mathbf{I}). \tag{7}$$

Due to the Markov property of the forward process, we have the marginal distribution of $\mathbf{x}_t$ given the initial clean data $\mathbf{x}_0$:

$$q(\mathbf{x}_t|\mathbf{x}_0) = \mathcal{N}(\mathbf{x}_t; \sqrt{\bar{\alpha}_t}\mathbf{x}_0, (1-\bar{\alpha}_t)\mathbf{I}), \tag{8}$$

where we denote $\alpha_t := 1 - \beta_t$ and $\bar{\alpha}_t := \prod_{s=1}^{t} \alpha_s$. Applying Bayes' rule, we can obtain the forward process posterior when conditioned on $\mathbf{x}_0$:

$$\begin{aligned} q(\mathbf{x}_{t-1}|\mathbf{x}_t, \mathbf{x}_0) &= \frac{q(\mathbf{x}_t|\mathbf{x}_{t-1}, \mathbf{x}_0)q(\mathbf{x}_{t-1}|\mathbf{x}_0)}{q(\mathbf{x}_t|\mathbf{x}_0)} \\ &= \frac{q(\mathbf{x}_t|\mathbf{x}_{t-1})q(\mathbf{x}_{t-1}|\mathbf{x}_0)}{q(\mathbf{x}_t|\mathbf{x}_0)}, \end{aligned} \tag{9}$$

where the second equation follows from the Markov property of the forward process. Since all three terms in Eq. 9 are Gaussians, the posterior $q(\mathbf{x}_{t-1}|\mathbf{x}_t, \mathbf{x}_0)$ is also a Gaussian distribution, and it can be written as

$$q(\mathbf{x}_{t-1}|\mathbf{x}_t, \mathbf{x}_0) = \mathcal{N}(\mathbf{x}_{t-1}; \tilde{\boldsymbol{\mu}}_t(\mathbf{x}_t, \mathbf{x}_0), \tilde{\beta}_t \mathbf{I}) \tag{10}$$

with mean $\tilde{\boldsymbol{\mu}}_t(\mathbf{x}_t, \mathbf{x}_0)$ and variance $\tilde{\beta}_t$. Plugging the expressions in Eq. 7 and Eq. 8 into Eq. 10, we obtain

$$\tilde{\boldsymbol{\mu}}_t(\mathbf{x}_t, \mathbf{x}_0) := \frac{\sqrt{\bar{\alpha}_{t-1}}\beta_t}{1-\bar{\alpha}_t}\mathbf{x}_0 + \frac{\sqrt{\alpha_t}(1-\bar{\alpha}_{t-1})}{1-\bar{\alpha}_t}\mathbf{x}_t \quad \text{and} \quad \tilde{\beta}_t := \frac{1-\bar{\alpha}_{t-1}}{1-\bar{\alpha}_t}\beta_t \tag{11}$$

after minor simplifications.

## B    PARAMETRIZATION OF DDPM

In Sec. 3.2, we mention that the parametrization of the denoising distribution for current diffusion models such as Ho et al. (2020) can be interpreted as $p_\theta(\mathbf{x}_{t-1}|\mathbf{x}_t) := q(\mathbf{x}_{t-1}|\mathbf{x}_t, \mathbf{x}_0 = f_\theta(\mathbf{x}_t, t))$. However, such a parametrization is not explicitly stated in Ho et al. (2020) but it is discussed by Song et al. (2021a). To avoid possible confusion, here we show that the parametrization of Ho et al. (2020) is equivalent to what we describe in Sec 3.2.

Ho et al. (2020) train a noise prediction network $\boldsymbol{\epsilon}_\theta(\mathbf{x}_t, t)$ which predicts the noise that perturbs data $\mathbf{x}_0$ to $\mathbf{x}_t$, and a sample from $p_\theta(\mathbf{x}_{t-1}|\mathbf{x}_t)$ is obtained as (see Algorithm 2 of Ho et al. (2020))

$$\mathbf{x}_{t-1} = \frac{1}{\sqrt{\alpha_t}}\left(\mathbf{x}_t - \frac{1-\alpha_t}{\sqrt{1-\bar{\alpha}_t}}\boldsymbol{\epsilon}_\theta(\mathbf{x}_t, t)\right) + \sigma_t \mathbf{z}, \tag{12}$$

where $\mathbf{z} \sim \mathcal{N}(\mathbf{0}, \mathbf{I})$ except for the last denoising step where $\mathbf{z} = 0$, and $\sigma_t = \sqrt{\tilde{\beta}_t}$ is the standard deviation of the Gaussian posterior distribution in Eq. 11.

Firstly, notice that predicting the perturbation noise $\boldsymbol{\epsilon}_\theta(\mathbf{x}_t, t)$ is equivalent to predicting $\mathbf{x}_0$. We know that $\mathbf{x}_t$ is generated by adding $\boldsymbol{\epsilon} \sim \mathcal{N}(\mathbf{0}, \mathbf{I})$ noise as:

$$\mathbf{x}_t = \sqrt{\bar{\alpha}_t}\mathbf{x}_0 + \sqrt{1-\bar{\alpha}_t}\boldsymbol{\epsilon},$$

Hence, after predicting the noise with $\boldsymbol{\epsilon}_\theta(\mathbf{x}_t, t)$ we can obtain a prediction of $\mathbf{x}_0$ using:

$$\mathbf{x}_0 = \frac{1}{\sqrt{\bar{\alpha}_t}}\left(\mathbf{x}_t - \sqrt{1-\bar{\alpha}_t}\boldsymbol{\epsilon}_\theta(\mathbf{x}_t, t)\right). \tag{13}$$

Next, we can plug the expression for $\mathbf{x}_0$ in Eq. 13 into the mean of the Gaussian posterior distribution in Eq. 11, and we have

$$\tilde{\boldsymbol{\mu}}_t(\mathbf{x}_t, \mathbf{x}_0) = \tilde{\mu}_t\left(\mathbf{x}_t, \frac{1}{\sqrt{\bar{\alpha}_t}}\left(\mathbf{x}_t - \sqrt{1 - \bar{\alpha}_t}\epsilon_\theta(\mathbf{x}_t, t)\right)\right) \tag{14}$$

$$= \frac{1}{\sqrt{\alpha_t}}\left(\mathbf{x}_t - \frac{1 - \alpha_t}{\sqrt{1 - \bar{\alpha}_t}}\epsilon_\theta(\mathbf{x}_t, t)\right) \tag{15}$$

after simplifications. Comparing this with Eq. 12, we observe that Eq. 12 simply corresponds to sampling from the Gaussian posterior distribution. Therefore, although Ho et al. (2020) use an alternative re-parametrization, their denoising distribution can still be equivalently interpreted as $p_\theta(\mathbf{x}_{t-1}|\mathbf{x}_t) := q(\mathbf{x}_{t-1}|\mathbf{x}_t, \mathbf{x}_0 = f_\theta(\mathbf{x}_t, t))$, i.e, first predicting $\mathbf{x}_0$ using the time-dependent denoising model, and then sampling $\mathbf{x}_{t-1}$ using the posterior distribution $q(\mathbf{x}_{t-1}|\mathbf{x}_t, \mathbf{x}_0)$ given $\mathbf{x}_t$ and the predicted $\mathbf{x}_0$.

## C  EXPERIMENTAL DETAILS

In this section, we present our experimental settings in detail.

### C.1  NETWORK STRUCTURE

**Generator:** Our generator structure largely follows the U-net structure (Ronneberger et al., 2015) used in NCSN++ (Song et al., 2021c), which consists of multiple ResNet blocks (He et al., 2016) and Attention blocks (Vaswani et al., 2017). Hyper-parameters for the network design, such as the number of blocks and number of channels, are reported in Table 6. We follow the default settings in Song et al. (2021c) for other network configurations not mentioned in the table, including Swish activation function, upsampling and downsampling with anti-aliasing based on Finite Impulse Response (FIR) (Zhang, 2019), re-scaling all skip connections by $\frac{1}{\sqrt{2}}$, using residual block design from BigGAN (Brock et al., 2018) and incorporating progressive growing architectures (Karras et al., 2020b). See Appendix H of Song et al. (2021c) for more details on these configurations.

We follow Ho et al. (2020) and use sinusoidal positional embeddings for conditioning on integer time steps. The dimension for the time embedding is $4\times$ the number of initial channels presented in Table 6. Contrary to previous works, we did not find the use of Dropout helpful in our case.

The fundamental difference between our generator network and the networks of previous diffusion models is that our generator takes an extra latent variable $\mathbf{z}$ as input. Inspired by the success of Style-GANs, we provide $\mathbf{z}$-conditioning to the NCSN++ architecture using *mapping networks*, introduced in StyleGAN (Karras et al., 2019). We use $\mathbf{z} \sim \mathcal{N}(\mathbf{0}, \mathbf{I})$ for all experiments. We replace all the group normalization (GN) layers in the network with adaptive group normalization (AdaGN) layers to allow the input of latent variables. The latent variable $\mathbf{z}$ is first transformed by a fully-connected network (called mapping network), and then the resulting embedding vector, denoted by $\mathbf{w}$, is sent to every AdaGN layer. Each AdaGN layer contains one fully-connected layer that takes $\mathbf{w}$ as input, and outputs the per-channel shift and scale parameters for the group normalization. The network's feature maps are then subject to affine transformations using these shift and scale parameters of the AdaGN layers. The mapping network and the fully-connected layer in AdaGN are independent of time steps $t$, as we found no extra benefit in incorporating time embeddings in these layers. Details about latent variables are also presented in Table 6.

**Discriminator:** We design our time-dependent discriminator with a convolutional network with ResNet blocks, where the design of the ResNet blocks is similar to that of the generator. The discriminator tries to discriminate real and fake $\mathbf{x}_{t-1}$, conditioned on $\mathbf{x}_t$ and $t$. The time conditioning is enforced by the same sinusoidal positional embedding as in the generator. The $\mathbf{x}_t$ conditioning is enforced by concatenating $\mathbf{x}_t$ and $\mathbf{x}_{t-1}$ as the input to the discriminator. We use LeakyReLU activations with a negative slope 0.2 for all layers. Similar to Karras et al. (2020b), we use a mini-batch standard deviation layer after all the ResNet blocks. We present the exact architecture of discriminators in Table 7.

Table 6: Hyper-parameters for the generator network.

| | CIFAR10 | CelebA-HQ | LSUN Church |
|---|---|---|---|
| # of ResNet blocks per scale | 2 | 2 | 2 |
| Initial # of channels | 128 | 64 | 64 |
| Channel multiplier for each scale | $(1, 2, 2, 2)$ | $(1, 1, 2, 2, 4, 4)$ | $(1, 1, 2, 2, 4, 4)$ |
| Scale of attention block | 16 | 16 | 16 |
| Latent Dimension | 100 | 100 | 100 |
| # of latent mapping layers | 3 | 3 | 3 |
| Latent embedding dimension | 256 | 256 | 256 |

Table 7: Network structures for the discriminator. The number on the right indicates the number of channels in each residual block.

| CIFAR-10 | CelebA-HQ and LSUN Church |
|---|---|
| | $1 \times 1$ conv2d, 128 |
| $1 \times 1$ conv2d, 128 | ResBlock down, 256 |
| ResBlock, 128 | ResBlock down, 512 |
| ResBlock down, 256 | ResBlock down, 512 |
| ResBlock down, 512 | ResBlock down, 512 |
| ResBlock down, 512 | ResBlock down, 512 |
| minibatch std layer | ResBlock down, 512 |
| Global Sum Pooling | minibatch std layer |
| FC layer $\rightarrow$ scalar | Global Sum Pooling |
| | FC layer $\rightarrow$ scalar |

## C.2 TRAINING

**Diffusion Process:** For all datasets, we set the number of diffusion steps to be 4. In order to compute $\beta_t$ per step, we use the discretization of the continuous-time extension of the process described in Eq. 1, which is called the Variance Preserving (VP) SDE by Song et al. (2021c). We compute $\beta_t$ based on the continuous-time diffusion model formulation, as it allows us to ensure that variance schedule stays the same independent of the number of diffusion steps. Let's define the normalized time variable by $t' := \frac{t}{T}$ which normalizes $t \in \{1, 2, \ldots, T\}$ to $[0, 1]$. The variance function of VP SDE is given by:

$$\sigma^2(t') = 1 - e^{-\beta_{\min} t' - 0.5(\beta_{\max} - \beta_{\min})t'^2},$$

with the constants $\beta_{\max} = 20$ and $\beta_{\min} = 0.1$. Recall that sampling from $t^{th}$ step in the forward diffusion process can be done with $q(\mathbf{x}_t | \mathbf{x}_0) = \mathcal{N}(\mathbf{x}_t; \sqrt{\bar{\alpha}_t}\mathbf{x}_0, (1 - \bar{\alpha}_t)\mathbf{I})$. We compute $\beta_t$ by solving $1 - \bar{\alpha}_t = \sigma^2(\frac{t}{T})$:

$$\beta_t = 1 - \alpha_t = 1 - \frac{\bar{\alpha}_t}{\bar{\alpha}_{t-1}} = 1 - \frac{1 - \sigma^2(\frac{t}{T})}{1 - \sigma^2(\frac{t-1}{T})} = 1 - e^{-\beta_{\min}(\frac{1}{T}) - 0.5(\beta_{\max} - \beta_{\min})\frac{2t-1}{T^2}}, \quad (16)$$

for $t \in \{1, 2, \ldots T\}$. This choice of $\beta_t$ values corresponds to equidistant steps in time according to VP SDE. Other choices are possible, but we did not explore them.

**Objective:** We train our denoising diffusion GAN with the following adversarial objective:

$$\min_{\phi} \sum_{t=1}^{T} \mathbb{E}_{q(\mathbf{x}_0)q(\mathbf{x}_{t-1}|\mathbf{x}_0)q(\mathbf{x}_t|\mathbf{x}_{t-1})} \left[ -\log(D_\phi(\mathbf{x}_{t-1}, \mathbf{x}_t, t) + \mathbb{E}_{p_\theta(\mathbf{x}_{t-1}|\mathbf{x}_t)}[-\log(1 - D_\phi(\mathbf{x}_{t-1}, \mathbf{x}_t, t))] \right]$$

$$\max_{\theta} \sum_{t=1}^{T} \mathbb{E}_{q(\mathbf{x}_0)q(\mathbf{x}_{t-1}|\mathbf{x}_0)q(\mathbf{x}_t|\mathbf{x}_{t-1})} \left[ \mathbb{E}_{p_\theta(\mathbf{x}_{t-1}|\mathbf{x}_t)}[\log(D_\phi(\mathbf{x}_{t-1}, \mathbf{x}_t, t))] \right]$$

where the outer expectation denotes ancestral sampling from $q(\mathbf{x}_0, \mathbf{x}_{t-1}, \mathbf{x}_t)$ and $p_\theta(\mathbf{x}_{t-1}|\mathbf{x}_t)$ is our implicit GAN denoising distribution.

Table 8: Optimization hyper-parameters.

|  | CIFAR10 | CelebA-HQ | LSUN Church |
|---|---|---|---|
| Initial learning rate for discriminator | $10^{-4}$ | $10^{-4}$ | $10^{-4}$ |
| Initial learning rate for generator | $1.6 \times 10^{-4}$ | $1.6 \times 10^{-4}$ | $2 \times 10^{-4}$ |
| Adam optimizer $\beta_1$ | 0.5 | 0.5 | 0.5 |
| Adam optimizer $\beta_2$ | 0.9 | 0.9 | 0.9 |
| EMA | 0.9999 | 0.999 | 0.999 |
| Batch size | 256 | 32 | 64 |
| # of training iterations | 400k | 750k | 600k |
| # of GPUs | 4 | 8 | 8 |

Similar to Ho et al. (2020), during training we randomly sample an integer time step $t \in [1, 2, 3, 4]$ for each datapoint in a batch. Besides the main objective, we also add an $R_1$ regularization term (Mescheder et al., 2018) to the objective for the discriminator. The $R_1$ term is defined as

$$R_1(\phi) = \frac{\gamma}{2} \mathbb{E}_{q(\mathbf{x}_0)q(\mathbf{x}_{t-1}|\mathbf{x}_0)q(\mathbf{x}_t|\mathbf{x}_{t-1})} \left[ \left\| \nabla_{\mathbf{x}_{t-1}} D_\phi(\mathbf{x}_{t-1}, \mathbf{x}_t, t) \right\|^2 \right],$$

where $\gamma$ is the coefficient for the regularization. We use $\gamma = 0.05$ for CIFAR-10, and $\gamma = 1$ for CelebA-HQ and LSUN Church. Note that the $R_1$ regularization is a gradient penalty that encourages the discriminator to stay smooth and improves the convergence of GAN training (Mescheder et al., 2018).

**Optimization:** We train our models using the Adam optimizer (Kingma & Ba, 2015). We use cosine learning rate decay (Loshchilov & Hutter, 2016) for training both the generator and discriminator. Similar to Ho et al. (2020); Song et al. (2021c); Karras et al. (2020a), we observe that applying an exponential moving average (EMA) on the generator is crucial to achieve high performance. We summarize the optimization hyper-parameters in Table 8.

We train our models on CIFAR-10 using 4 V100 GPUs. On CelebA-HQ and LSUN Church we use 8 V100 GPUs. The training takes approximately 48 hours on CIFAR-10, and 180 hours on CelebA-HQ and LSUN Church.

### C.3    EVALUATION

When evaluating IS, FID and recall score, we use 50k generated samples for CIFAR-10 and LSUN Church, and 30k samples for CelebA-HQ (since the CelebA HQ dataset contains only 30k samples).

When evaluating sampling time, we use models trained on CIFAR-10 and generate a batch of 100 samples. We benchmark the sampling time on a machine with a single V100 GPU. We use Pytorch 1.9.0 and CUDA 11.0.

### C.4    ABLATION STUDIES

Here we introduce the settings for the ablation study in Sec. 5.2. We observe that training requires a larger number of training iterations when $T$ is larger. As a result, we train the model for each $T$ until the FID score does not increase any further. The number of training iteration is 200k for $T = 1$ and $T = 2$, 400k for $T = 4$ and 600k for $T = 8$. We use the same network structures and optimization settings as in the main experiments.

For the data augmentation baseline, we follow the differentiable data augmentation pipeline in Zhao et al. (2020). In particular, for every (real or fake) image in the batch, we perturbed it by sampling from a random timestep at the diffusion process (except the last diffusion step where the information of data is completely destroyed). We find the results insensitive to the number of possible perturbation levels (i.e, the number of steps in the diffusion process), and we report the result using a diffusion process with 4 steps. Since the perturbation by the diffusion process is differentiable due to the re-parametrization trick (Kingma & Welling, 2014), we can train both the discriminator and generator with the perturbed samples. See Zhao et al. (2020) for a detailed explanation for the training pipeline.

For the experiments on alternative parametrizations, we use $T = 4$ for the diffusion process and keep other settings the same as in the main experiments.

For the experiment on training a model without latent variables, similar to the main experiments, the generator takes the conditioning $\mathbf{x}_t$ as its input, and the time conditioning is still enforced by the time embedding. However, the AdaGN layers are replaced by plain GN layers, such that no latent variable is needed, and the mapping network for $\mathbf{z}$ is removed. Other settings follow the main experiments.

### C.5 TOY DATA AND STACKEDMNIST

For the 25-Gaussian toy dataset, both our generator and discriminator have 3 fully-connected layers each with 512 hidden units and LeakyReLU activations (negative slope of 0.2). We enforce both the conditioning on $\mathbf{x}_t$ and $t$ by concatenation with the input. We use the Adam optimizer with a learning rate of $10^{-4}$ for both the generator and discriminator. The batch size is 512, and we train the model for 50k iterations.

Our experimental settings for StackedMNIST are the same as those for CIFAR-10, except that we train the model for only 150k iterations.

## D TRAINING STABILITY

In Fig. 10, we plot the discriminator loss for different time steps in the diffusion process when $T = 4$. We observe that the training of our denoising diffusion GAN is stable and we do not see any explosion in loss values, as is sometimes reported for other GAN methods such as Brock et al. (2018). The stability might be attributed to two reasons: First, the conditioning on $\mathbf{x}_t$ for both generator and discriminator provides a strong signal. The generator is required to generate a few plausible samples given $\mathbf{x}_t$ and the discriminator requires classifying them. The $\mathbf{x}_t$ conditioning keeps the discriminator and generator in a balance. Second, we are training the GAN on relatively smooth distributions, as the diffusion process is known as a smoothening process that brings the distributions of fake and real samples closer to each other (Lyu, 2012). As we can see from Fig. 10, the discriminator loss for $t > 0$ is higher than $t = 0$ (the last denoising step). Note that $t > 0$ corresponds to training the discriminator on noisy images, and in this case the true and generator distributions are closer to each other, making the discrimination harder and hence resulting in higher discriminator loss. We believe that such a property prevents the discriminator from overfitting, which leads to better training stability.

## E ADDITIONAL QUALITATIVE RESULTS

We show additional qualitative samples of CIFAR-10, CelebA-HQ and LSUN Church Outdoor in Figure 11, Figure 12 and Figure 13, respectively.

## F ADDITIONAL VISUALIZATION FOR $p_\theta(\mathbf{x}_0|\mathbf{x}_t)$

In Figure 14 and Figure 15, we show visualizations of samples from $p_\theta(\mathbf{x}_0|\mathbf{x}_t)$ for different $t$. Note that except for $p_\theta(\mathbf{x}_0|\mathbf{x}_1)$, the samples from $p_\theta(\mathbf{x}_0|\mathbf{x}_t)$ do not need to be sharp, as they are only intermediate outputs of the sampling process. The conditioning is less preserved as the perturbation in $\mathbf{x}_t$ increases, and in particular $\mathbf{x}_T$ ($\mathbf{x}_4$ in our example) contains almost no information of clean data $\mathbf{x}_0$.

## G NEAREST NEIGHBOR RESULTS

In Figure 16 and Figure 17, we show the nearest neighbors in the training dataset, corresponding to a few generated samples, where the nearest neighbors are computed using the feature distance of a pre-trained VGG network (Simonyan & Zisserman, 2014). We observe that the nearest neighbors are significantly different from the samples, suggesting that our models generalize well.

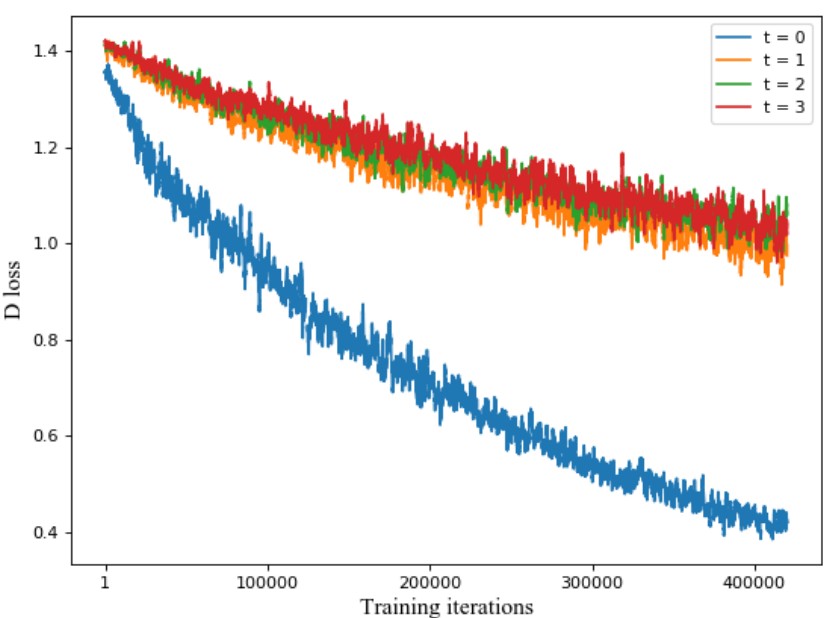

Figure 10: The discriminator loss per denoising step during training.

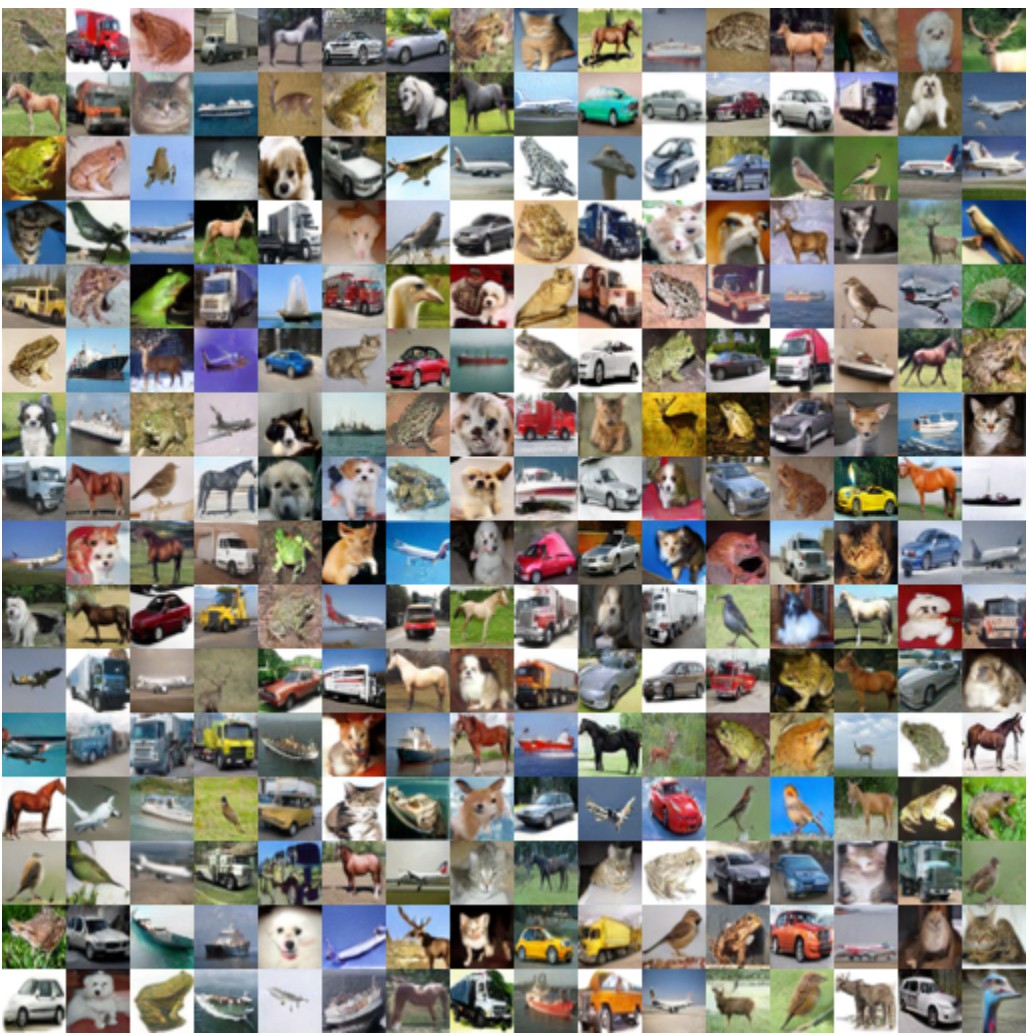

Figure 11: Additional qualitative samples on CIFAR-10.

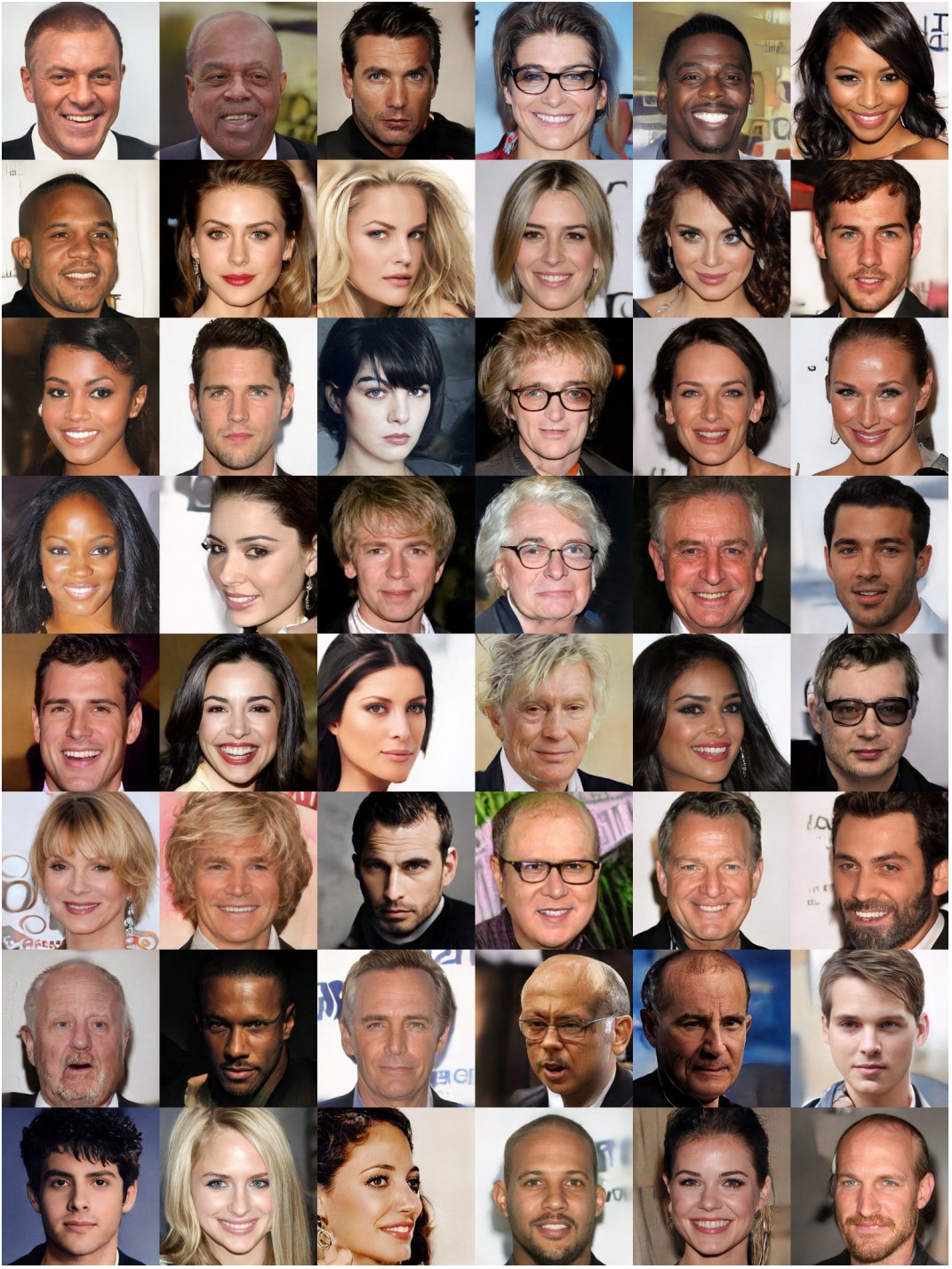

Figure 12: Additional qualitative samples on CelebA-HQ.

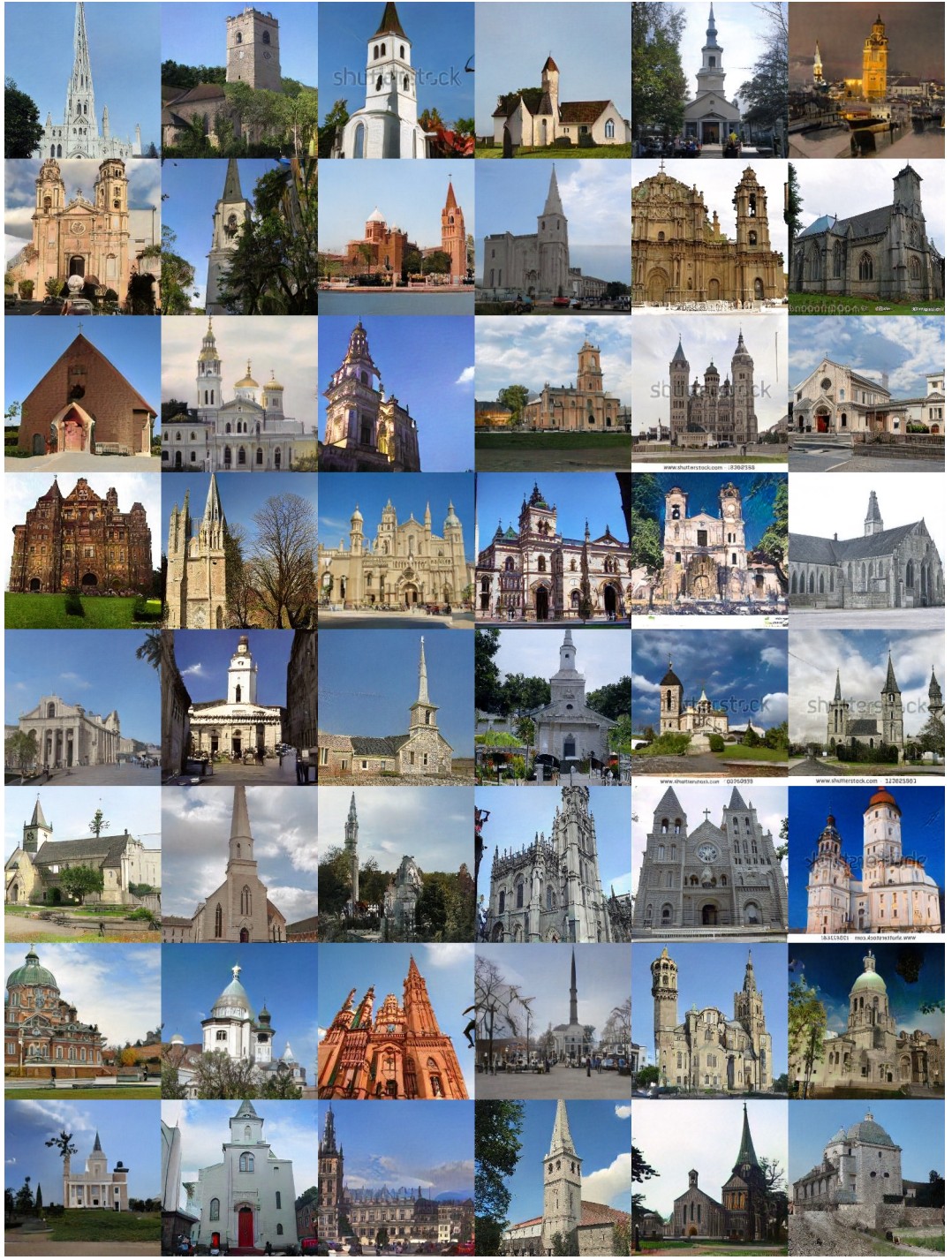

Figure 13: Additional qualitative samples on LSUN Church Outdoor.

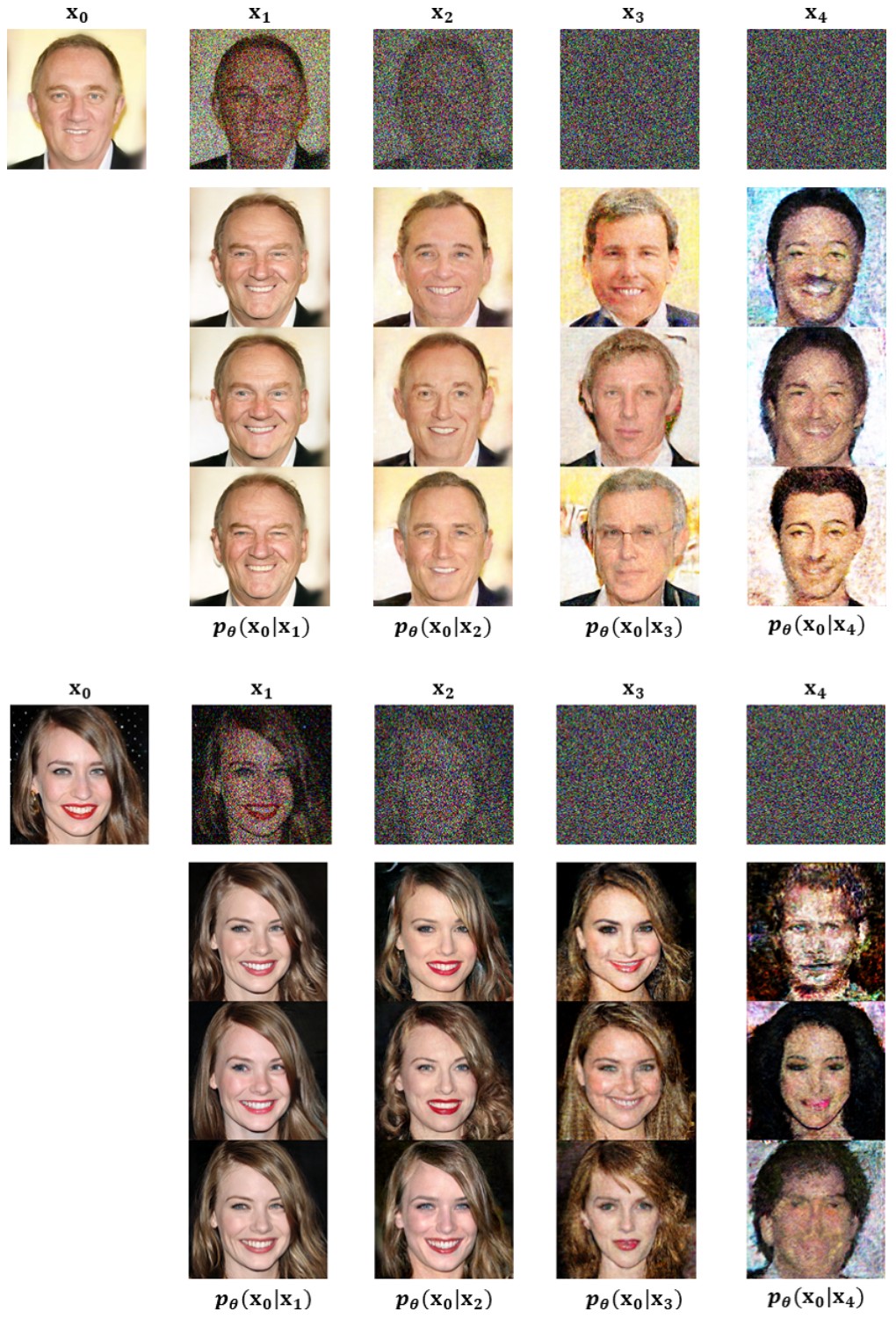

Figure 14: Visualization of samples from $p_\theta(\mathbf{x}_0|\mathbf{x}_t)$ for different $t$ on CelebA-HQ. For each example, the top row contains $\mathbf{x}_t$ from diffusion process steps, where $\mathbf{x}_0$ is a sample from the dataset. The bottom rows contain 3 samples from $p_\theta(\mathbf{x}_0|\mathbf{x}_t)$ for different $t$'s.

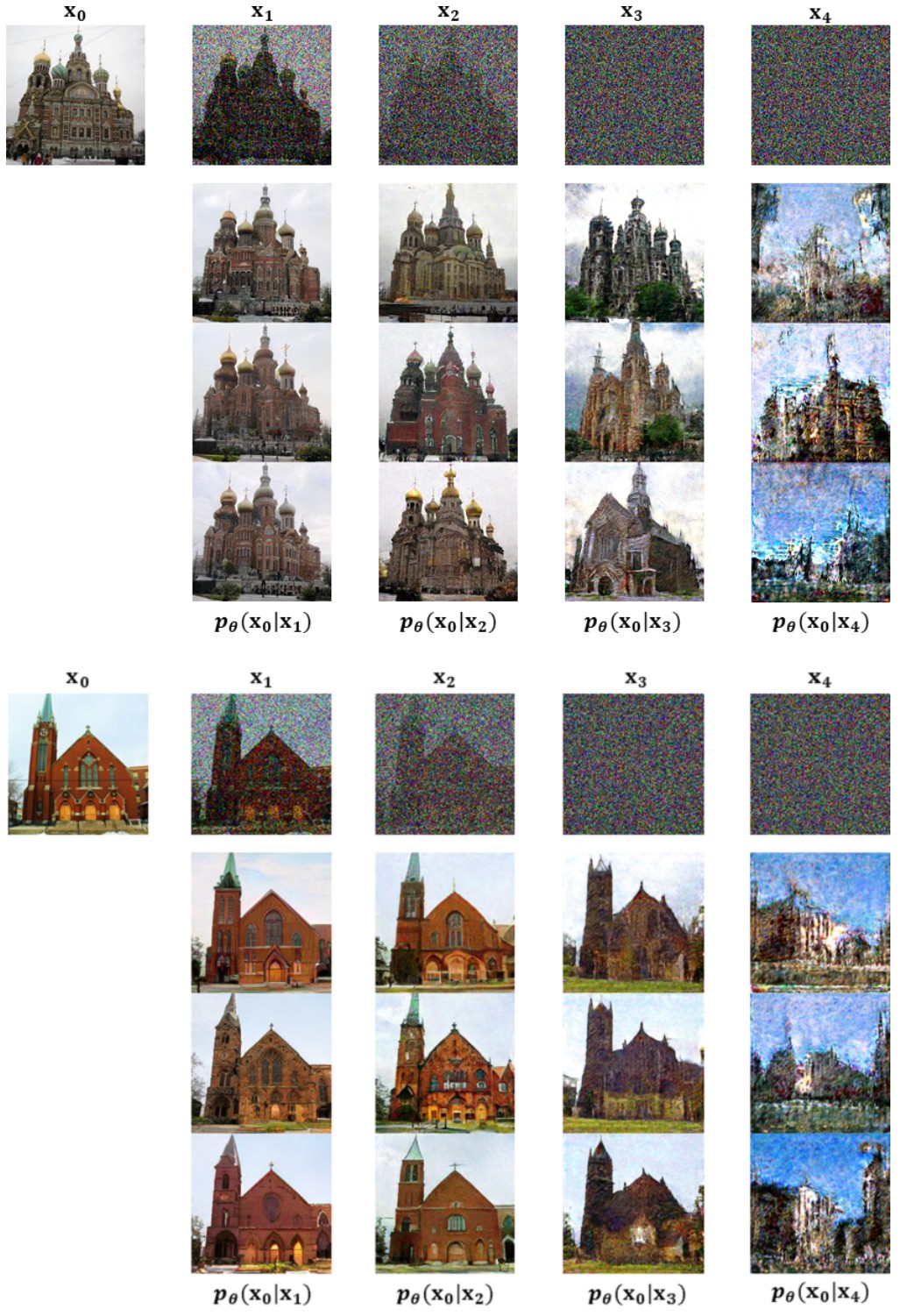

Figure 15: Visualization of samples from $p_\theta(\mathbf{x}_0|\mathbf{x}_t)$ for different $t$ on LSUN Church. For each example, the top row contains $\mathbf{x}_t$ from diffusion process steps, where $\mathbf{x}_0$ is a sample from the dataset. The bottom rows contain 3 samples from $p_\theta(\mathbf{x}_0|\mathbf{x}_t)$ for different $t$'s.

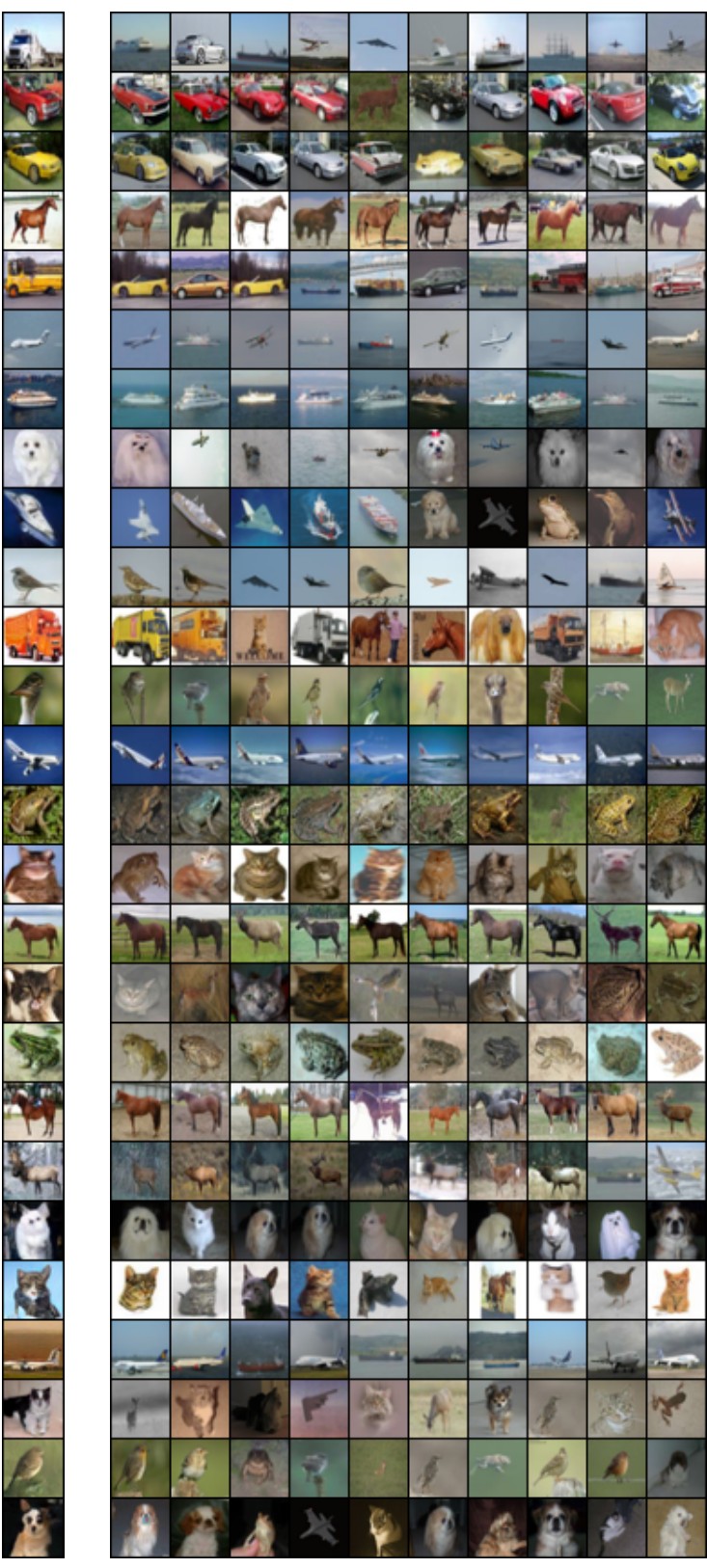

Figure 16: CIFAR-10 nearest neighbors in VGG feature space. Generated samples are in the leftmost column, and training set nearest neighbors are in the remaining columns.

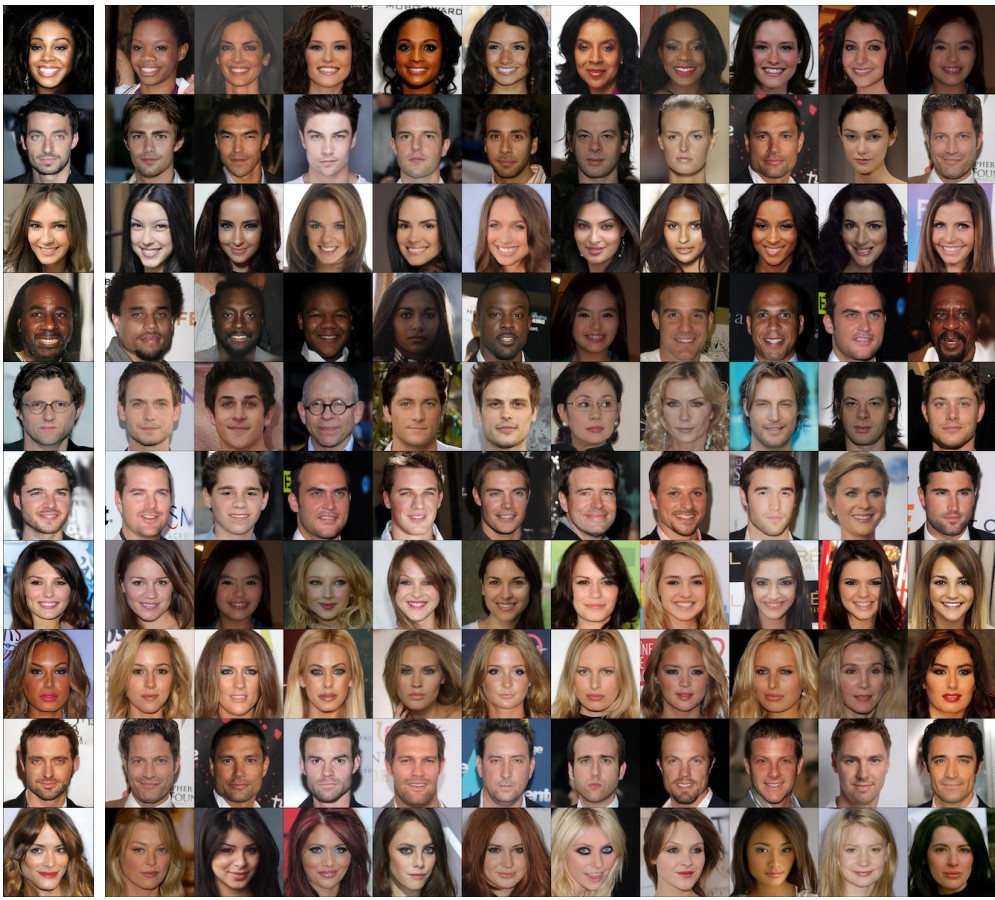

Figure 17: CelebA-HQ nearest neighbors in the VGG feature space. Generated samples are in the leftmost column, and training set nearest neighbors are in the remaining columns.

