# OpenReview forum: "Tackling the Generative Learning Trilemma with Denoising Diffusion GANs"
_ICLR.cc/2022/Conference — ICLR 2022 Spotlight_

### Official Review · Reviewer_FKLx · 2021-11-02

**Correctness:** 4
**Technical Novelty And Significance:** 4
**Empirical Novelty And Significance:** 4
**Recommendation:** 8
**Confidence:** 3

**Main Review:**

The authors start by showing with a toy example that the denoising conditional
distribution can be heavily multi-modal for large diffusion steps, highlighting
the limitations of the typical Gaussian model, which can only be assumed for
very small steps, thus leading to an expensive multi-step sampling process. The
authors overcome this by using a conditional GAN to model the denoising
distribution under large steps, and experimentally show that this can indeed
model very multimodal distributions. This allows the reversal of the diffusion
process using much fewer steps than in competing methods, achieving efficient
sampling speeds while maintaining the high sample quality and mode coverage of
diffusion models. While taking this idea to the extreme would lead to the
standard one-step GAN, interestingly the authors show that their method
outperforms standard GANs, and the provided justification is that in the
multi-step approach the distribution mapping in intermediate steps is less
aggressive than direct Gaussian-to-data-distribution, and thus easier to learn.

To the best of my understanding I couldn't identify any serious weakness in
this work.

**Summary Of The Paper:**

This paper proposes to use a conditional GAN as a more expressive model of the
denoising step in a denoising diffusion model, allowing to reverse the diffusion
(sampling) much more efficiently, while maintaining the high quality of diffusion
models.

**Summary Of The Review:**

I think the proposed idea is clever and very well executed. Experimental results
validate the method in terms of the triple sample quality / mode coverage /
sampling efficiency trade-off. Experiments demonstrate their conditional
generator exhibits the strong multi-modality required for the large reverse
diffusion steps.  I expect this hybrid between conditional GANs and diffusion
models to become an important player in generative modeling.

---

> ### Author Response · Authors · 2021-11-12
> **Thanks for the positive feedback!**
>
> We thank the reviewer for the positive feedback. We share the same belief that denoising diffusion GANs will be important players in the generative learning arena. Given their simplicity and efficacy in tackling the generative learning trilemma, we believe they will be widely used by the community.

---

### Official Review · Reviewer_27CK · 2021-11-02

**Correctness:** 4
**Technical Novelty And Significance:** 3
**Empirical Novelty And Significance:** 3
**Recommendation:** 8
**Confidence:** 4

**Main Review:**

**Strengths**

I think this work and the proposed combination of adversarial and likelihood-based training is a natural and novel extension of DDPMs towards faster sampling times. The main idea of modeling the conditional transition probabilities with more informative models is simple (which is a good thing). Most design decisions are justified by ablation experiments (e.g., type of parameterization, number of diffusion steps, deterministic vs. stochastic generators, etc.). Experiments with the 25-Gaussian and StackedMNIST datasets show the improved mode coverage over GAN models.

The results on CIFAR-10, CelebA-HQ and LSUN Churches are competitive with previous state-of-the-art methods. Furthermore, I really like the underlying motivation of the work that interactive image processing or speech synthesis require very fast sampling, and I agree that this is a pressing issue when dealing with diffusion models.

**Weaknesses**

- Fig. 1 is a good motivator, but a bit imprecise: (non-reweighted) diffusion models also belong to the class of likelihood-based models, as do autoregressive generative models. The latter are slow, but cover the modes and provide high quality.
- How important is the choice of variance schedule? If I understand correctly, the work relies exclusively on the schedule given in equation (16). How does the schedule affect controlled synthesis, e.g., stroke-based image synthesis?
- The paper should address previous work that also attempts to solve the trilemma by reducing the number of steps required for a trained DDPM model, e.g., DDIM [1] or GGF [2], and compare with them.
- It should also be shown that the coverage of modes on more complex, high-resolution data (such as LSUN Churches) is still as good as the results on the Toy data suggest.
- Do the results presented in Tab. 2 (ablation studies on CIFAR-10) also hold for high-resolution (i.e. $256 \times 256$) datasets?

**Mixed Comments:**

- Is there evidence that diffusion models actually provide (approximate) SOTA diversity? My impression is that they can be tuned either to very good likelihoods (as in [3]) or to high quality samples (as in [4]), but currently cannot do both at the same time. The best performing diffusion models in terms of _quality_ are based on the reweighted objective introduced in [5], which sacrifices likelihood interpretation for better sample quality.
- Figure 9 nicely demonstrates controlled image synthesis/modification. However, compared to [6] only 4 steps provide less fine-grained control over how much content is obtained.
- How does the model perform on other typical but more complex synthesis tasks such as LSUN Cats or class-conditional generation of ImageNet?
- Just out of pure interest: Have you tried using the UNet as a discriminator as well (similar to [7])? How important is the exact implementation of the discriminator architecture?

__References__

- [1]: Song, J., Meng, C., Ermon, S.: Denoising Diffusion Implicit Models
- [2]: Jolicoeur-Martineau, Alexia, et al. "Gotta Go Fast When Generating Data with Score-Based Models."
- [3]: Kingma, Diederik P., et al. "Variational diffusion models."
- [4]: Dhariwal, Prafulla, and Alex Nichol. "Diffusion models beat gans on image synthesis."
- [5]: Ho, Jonathan, Ajay Jain, and Pieter Abbeel. "Denoising diffusion probabilistic models."
- [6]: Meng, Chenlin, et al. "Sdedit: Image synthesis and editing with stochastic differential equations."
- [7]:  Schonfeld, Edgar, Bernt Schiele, and Anna Khoreva. "A u-net based discriminator for generative adversarial networks."


**Summary Of The Paper:**

This paper introduces _Denoising Diffusion GANs (DDGANs)_ to solve the so-called "generative learning trilemma" by training a generative model that achieves high-quality sampling, fast sampling times, and at the same time, high mode coverage. The main intuition of the paper is to combine GANs, which have been shown to achieve both high-quality synthesis and very fast sampling times, and DDPMs, which provide high-quality sampling and good mode coverage. In contrast to previous work on diffusion models that rely on Gaussian transition probabilities of the reverse forward process, conditional GANs are used in this work to allow flexible, multimodal denoising distributions in the reverse process and thus much shorter sequence lengths. Experiments show that the proposed approach indeed achieves higher mode coverage than conventional GAN models while maintaining high sample quality within 4 to 8 diffusion steps. The model can also be used for stroke-based image synthesis, with significant improvements in sampling times compared to previous approaches.

**Summary Of The Review:**

In summary, the paper provides a nice combination of adversarial and likelihood-based training and, in my opinion, represents a natural and novel extension of DDPMs towards faster sampling times. The experiments validate the claim that denoising diffusion GANs can indeed help overcome the "generative learning trilemma", and most of the ablation studies justify the design choices made in the paper. I think this is a good paper worth publishing at ICLR 2022, and I am willing to increase my score if my remaining questions and concerns are adequately answered.

---

> ### Author Response · Authors · 2021-11-12
> **Thanks for the positive feed back and detailed responses**
>
> Thank you for providing detailed feedback. **We hope that our response below can address your main concerns. If so, we would appreciate it if you considered raising your score for this submission.** If you have any further thoughts or questions, please feel free to get back to us in the discussion period.
>
> **Updates to Figure 1**:
> We thank the reviewer for pointing this out. We agree that diffusion models also belong to the likelihood-based family. To avoid possible confusion, in the revised version we remove the term “Likelihood-based Models” in Figure 1 and we only list variational autoencoders and normalizing flows as models that satisfy fast sampling and mode coverage.
>
>
> **Variance Schedule**:
> To keep things simple, we follow the original DDPM [1] paper to choose the variance schedule. In particular, the variance obtained from Eq. 16 for each step gives exactly the same variance as in [1] for the forward diffusion process (for $q(x_t|x_0)$). Our choice of $\beta_{min}$ and $\beta_{max}$ also follows [1]. Eq. 16 is derived from [2] that connects the discrete-time diffusion process in [1] to the continuous-time variance-preserving SDE (VP-SDE). We express the variance scheduling using Eq. 16, because it conveniently computes variances for different time-steps from normalized time values (i.e., $\frac{t}{T}$).
>
> We use the variance schedule from the original DDPM for all experiments, including the stroke-based image synthesis. Thus, our strong results do not come from tuning the schedule. We believe that an interesting future direction is to explore alternative schedules, designed specifically for a small number of timesteps in the denoising diffusion GAN framework.
>
> **Previous work (DDIM and GGF) on accelerating sampling from diffusion models**:
> We would like to kindly remind you that we indeed study previous approaches including DDIM and GGF that accelerate sampling from denoising diffusion models. In particular, in Table 1, we report the number of function evaluations and sampling time for DDIM, GGF, as well as several other competitive methods in this space, such as FastDDPM [3] and LSGM [4]. In Figure 4, we compare our model to these methods in terms of sampling time and sample quality. From both Table 1 and Figure 4, we observe that our method is in a much stronger position in the sampling time and quality trade-off compared to these methods that aim to accelerate sampling from diffusion models.
>
> **Mode coverage on other datasets**:
> In addition to the toy dataset, we study the mode coverage of our model on the StackedMNIST dataset (Table 3) and we report the recall score on CIFAR-10 (Table 1). From these results, we observe that the mode coverage of our denoising diffusion GAN is similar to likelihood-based and diffusion models, and significantly better than GANs (even compared to those GANs that are specifically designed to alleviate mode collapse, as shown in Table 3). We believe that the current results provide strong evidence to support the claim that our model provides good mode coverage. We will be happy to run the recall score analysis of our model for high dimensional image datasets such as LSUN, if the reviewer believes that it is crucial for the acceptance of this paper. However, please note that obtaining a baseline recall score for previous diffusion-based models on these datasets is extremely expensive. Generating a batch of 16 images from a model provided in [2] using PC sampling requires 45 minutes on a single GPU according to [4]. This translates to 2300 GPU hours to generate 50K samples from this model.
>
>
> **Rerunning ablation studies on larger images**:
> We have done the ablation for one-shot unconditional GAN ($T=1$) on larger images. In particular, for $T=1$, we have FID 16.7 on CelebA-HQ, and 14.3 on LSUN Church, which are both significantly worse than our denoising diffusion GAN results. Due to the high computational cost of running experiments on large image datasets, we do not do ablation studies on them. Please note that we are following the common practice in which we perform extensive ablation studies on relatively small datasets (CIFAR-10 in our case) to obtain insights on our model, and we follow the same insights in other experiments.
>
>
> [1] Ho et al., Denoising Diffusion Probabilistic Models
>
> [2] Song et al., Score-Based Generative Modeling through Stochastic Differential Equations
>
> [3] Kong and Ping, On Fast Sampling of Diffusion Probabilistic Models
>
> [4] Vahdat et al., Score-based Generative Modeling in Latent Space

---

> > ### Author Response · Authors · 2021-11-12
> > **Continue**
> >
> > **Likelihood-FID trade-off in diffusion models**:
> > We agree that in diffusion models, the best FID models and the best likelihood models are often tuned differently. However, even denoising diffusion models tuned for FID scores often obtain relatively good test likelihood. For example, Song et al. [2] obtain FID 2.92 and NLL 2.99 on CIFAR-10. This indicates that the diffusion models do not miss modes, as missing a mode would result in very high negative log-likelihood (or even infinite values). That is why we believe that even an FID-tuned diffusion model can serve as a strong model for diversity.
> >
> > **Fine-grained control on image synthesis/editing**:
> > Since our model only has 4 discrete time steps, it has a limited choice for the amount of perturbation when performing controlled image synthesis and editing tasks. Therefore, we agree that our model provides less fine-grained control over the magnitude of perturbations in these tasks. However, in [5] the authors only consider perturbing the data to $t = 0.5$, which is also what we do in our experiments. Furthermore, it is worth noting that having less steps, like in our denoising diffusion GAN framework, does generally not imply less expressivity in the controlled synthesis/edit operation, because our generator defines an expressive, multimodal distribution---in contrast to a single step in a standard denoising diffusion model.
> >
> > Finally, note that our focus is not image editing, and we provide experimental results for this task as a proof of concept to show how our model can significantly accelerate sampling in downstream applications of diffusion models. However, we believe that real-time controlled synthesis and image editing could be an important application of denoising diffusion GANs and we hope that our work stimulates further research into this direction.
> >
> >
> > **Performance on other datasets**:
> > We currently study sample quality on CIFAR-10, LSUN Church outdoor, and CelebA-HQ. Note that LSUN Church outdoor is a fairly complex high-dimensional image dataset with diverse samples. While it is definitely interesting to explore more challenging tasks such as class conditional image generation on ImageNet, we leave the study of denoising diffusion GANs on these tasks to future works.
> >
> > **U-net discriminator**:
> > We thank the reviewer for pointing out this work. The idea of using a U-net for the discriminator is an interesting direction to explore. To keep the framework simple, we currently rely on common convolutional discriminators as discussed in Appendix C. However, we believe our results could be further improved with a better generator/discriminator architecture design, as you pointed out.
> >
> > [5] Meng et al. Sdedit: Image synthesis and editing with stochastic differential equations.

---

> > > ### Comment · Reviewer_27CK · 2021-11-18
> > > **Thanks!**
> > >
> > > Thanks for the clarifications and apologies for overlooking the ablations regarding the fast samplers.
> > >
> > > I'm remaining curious to see how the approach performs on more complex problems, and look forward to experimenting with this model myself. Raising the score to 8.

---

### Official Review · Reviewer_6zYL · 2021-11-03

**Correctness:** 4
**Technical Novelty And Significance:** 3
**Empirical Novelty And Significance:** 4
**Recommendation:** 8
**Confidence:** 4

**Main Review:**

Presentation
- The paper is easy to read and understand. It contains many helpful figures, qualitative examples and quantitative comparisons summarized in tables. I especially like Fig. 14 which is very useful in understanding the amount of information within each of the diffusion scales, since this is sometimes very difficult to judge from a visualization of the noisy states themselves.
- The paper motivates the problem it is addressing well through the mental image of a trilemma.

Literature/Background
- The related work section covers many relevant works and provides a good classification of the current work within the existing literature.
- It contains the necessary background material on diffusion models to be sufficiently self-contained.

Approach and evaluation
- The approach is relatively simple and provides good results. Experiments convincingly demonstrate that the approach indeed strikes a good balance of the three desired goals.
- A number of ablations analyze different aspects of the model in more detail.
- Besides helping with mode coverage, the approach brings other advantages over a pure GAN training such as discriminator regularization helping against overfitting on small datasets, and potentially more control over the generation process (as also evidenced by the stroke-based synthesis).

Additional Comments
- Loss weighting: For diffusion models, the weight applied to the objective from different steps of the diffusion process can be critical. In fact, a change in this weighting has been one of the break-throughs in [Ho, 2020] that lead to the current success of this class of models. The choice of this weight in the presented approach is not discussed. Is it simply the same for all timesteps? Did you experiment with this?
- Choice of generator architecture: While it is nice to see that a similar model as used for diffusion models also works as the conditional generator in the presented setting, Tab. 2 suggests that it is actually a bad choice for an unconditional/one-step generator. Have you experimented with different choices there? Does it help to use a different architecture for the first (unconditional) step compared to the other steps?

**Summary Of The Paper:**

The goal of the paper is to develop a generative model which achieves three desirable goals at the same time: High-quality samples, data/mode coverage and fast sampling times. While GANs have been associated with high-quality samples and fast sampling times, they are also known for their tendency to ignore modes of the data. On the other hand, diffusion models have recently been demonstrated to achieve high-quality samples and good likelihood estimation, which can indicate good mode coverage. However, diffusion models tend to be slow to generate samples due to their iterative nature. Thus, the paper proposes to combine diffusion models with GANs to combine their complementary advantages regarding mode coverage and sampling times (together with their shared ability to produce high-quality samples).

The main reason why diffusion models require a lot of iterations is because they model each of the steps of the reverse diffusion process with a Gaussian distribution, which is only valid for very small timesteps/a large number of steps. Thus, instead of using a network to estimate parameters of a Gaussian reverse process, the authors propose to model steps of the reverse process with a GAN. In theory, a GAN could directly model the data distribution and therefore solve the task in a single step, where the approach would be equivalent to a GAN and therefore suffer again from mode collapse. The main hope is therefore that by using a few diffusion steps, such an approach is still relatively fast compared to pure diffusion models while also having a good mode coverage.

Experiments evaluate sample quality, sampling speed and mode coverage. The main evaluation on CIFAR-10 indeed demonstrates a favorable combination of sample quality according to FID and IS, sampling speed and data coverage as measured by Recall. Additional experiments analyze the effect of various ablations and demonstrate an application in stroke-based image synthesis.

**Summary Of The Review:**

Diffusion models come with a lot of desired benefits such as high quality samples, data coverage and stable training. However, their slow sampling speeds limit their applicability and the problem of improving this aspect with a new generative model is very relevant. The idea of combining diffusion models with the benefits of GANs by modeling the reverse process with conditional GANs is plausible. The main question is whether such an approach can really combine the advantages of the different models. To that end, the experiments are quite convincing and I think the paper provides a positive answer to that question, which will make it significant for future research. Thus, I recommend acceptance.

---

> ### Author Response · Authors · 2021-11-12
> **Thanks for the positive feedback!**
>
> We thank the reviewer for providing positive feedback and recognizing the significance of this work. Here we will address two main questions.
>
> **Loss weighting**:
> In all our experiments, we weight the loss for different timesteps equally and we do not introduce any reweighting as shown in Eq. 4. Previous works on diffusion models [1, 2, 3, 4] are derived from a likelihood-based criterion that emphasizes heavily low-level image statistics. As you pointed out, these works report that reweighting the objective can trade likelihood for image quality. However, our objective is not derived from a likelihood perspective and it uses an extremely small number of timesteps. It is not trivially clear how a heuristically designed reweighting mechanism can be defined for our case.
>
> We examined a few heuristic weighting schemes in our early experiments, but we did not observe any significant improvements in the sample quality. Therefore, for the sake of simplicity, we do not introduce any reweighting in our training.
>
> **Choice of generator architecture**:
> We thank the reviewer for raising this interesting question. Although our generator structure does reasonably well for $T=1$, the performance for this case indeed is worse than the state-of-the-art GANs, suggesting that the architecture may not be optimal for an unconditional generation. We adopt the U-Net structure mainly to accommodate the conditioning on $x_t$, and to have a fair comparison with prior diffusion models that use the same architecture. Our primary goal in this paper is to focus on improvements that can be obtained solely from our training formulation (instead of architecture improvements). However, we share the same belief that denoising diffusion GANs can be further improved with better network architectures (especially for the first sampling step).
>
>
> [1] Ho et al., Denoising Diffusion Probabilistic Models
>
> [2] Song et al., Maximum Likelihood Training of Score-Based Diffusion Models.
>
> [3] Kingma et al., Variational Diffusion Models
>
> [4] Vahdat et al., Score-based Generative Modeling in Latent Space

---

### Official Review · Reviewer_XcPr · 2021-11-04

**Correctness:** 4
**Technical Novelty And Significance:** 3
**Empirical Novelty And Significance:** 3
**Recommendation:** 8
**Confidence:** 4

**Main Review:**

Generally, this paper proposes a good idea to solve the trilemma in generative learning, which accelerating the sampling in denoising diffusion models by using conditional GANs. The idea is clean with the experimental support. There are several minor issues in the required addressing as listed below.

1) It is desirable if the conditional GANs in the proposed model can be tested with different settings. It is currently non-saturating GANs. Does the proposed approach also work better with advanced conditional GANs?
2) The results of FID should be specified with the evaluation dataset or both (FID (train) and FID (test)).
3) It is desirable to make the diffusion model continuously improve as the iteration increases. However, when T is greater than 4, the performance decreases.
4) The format of references should be consistent (e.g., with or without abbreviations).

**Summary Of The Paper:**

The diffusion models or score matching-based models have achieved great success in generating high fidelity samples with diverse modes. However, the sampling requires tremendous iterations, which hinders the practical usages. This paper proposes to break the Gaussian assumption in denoising steps by parametrizing it with a multimodal conditional GAN. The time for sampling can be 2000 times faster than the original diffusion model while keeping competitive quality and diversity.

**Summary Of The Review:**

This paper solves an important problem in denoising diffusion models by parametrizing denoising with a multimodal conditional GAN. Experimental results show that the proposed model achieves good sample quality, mode coverage and speed without struggling in the trilemma. The minor issues are in the experiments. It is encouraged to test and discuss different setting. Therefore, I recommend to accept this paper.

---

> ### Author Response · Authors · 2021-11-12
> **Thanks for the positive feedback!**
>
> We thank the reviewer for recognizing the impact of this work. Below, we address the comments.
>
> **Different GAN objectives**:
> We currently opt for the non-saturated GAN loss mainly because of its simplicity and its success in state-of-the-art GAN frameworks such as StyleGANs. In the early stages of the project, we examined alternative objectives, including least-square GAN (LSGAN) and Wasserstein GAN (W-GAN) losses. We found that LSGAN leads to performance similar to the non-saturated loss, while W-GAN results in a slightly worse sampling quality. It is possible that extra hyper-parameter tuning is required to train denoising diffusion GANs with these objectives. However, for the sake of simplicity, we do not include the results of other adversarial losses in our paper, and we leave the detailed study of such losses to future works.
>
> **Specification of FID (train) and FID (test)**:
> We thank the reviewer for pointing this out. Throughout the paper, we follow the common approach of using FID (train) for evaluation. In the revised version, we now explicitly state this in Section 5.1.
>
> **Decrease in performance when $T$ is larger**:
> We agree that diffusion models perform better when the number of steps is larger. In our formulation, the denoising model is trained using adversarial loss, while in current denoising diffusion models, this is done by minimizing an $L_2$ loss. We hypothesize that several factors may make training our model for larger $T$’s more challenging: i) In the adversarial setting, we rely on the discriminator to measure the mismatch between the parametric denoising model and true denoising distribution. The discriminator may not be as effective as the $L_2$ loss when $T$ gets larger, in which the true denoising distribution becomes more Gaussian. ii) Both generator and discriminator may require more capacity as $T$ gets larger. In particular, the generator, shared across all timesteps, may require more capacity to denoise images to their corresponding fine-grained denoised versions, and the discriminator is tasked with distinguishing increasingly more similar real/fake samples. Overall, we think it is plausible that the adversarial objective of denoising diffusion GANs may behave differently with respect to the number of steps $T$, compared to the simpler regression-like objectives of standard diffusion models.
>
> We think it will be interesting future work to study tradeoffs in designing an adversarial training setup with a larger number of timesteps. We believe generators and discriminators designed with correct inductive biases for diffusion models may further improve the performance of denoising diffusion GANs.
>
> **Citation format**:
> We thank the reviewer for pointing this out. In the revised version, we have fixed the citations following your advice. Please let us know if they require further improvements.

---

### Decision · Program_Chairs · 2022-01-20

**Decision:**

Accept (Spotlight)

**Comment:**

The paper modifies DPMs by replacing the denoising L2 losses with GANs to learn the iterative denoising process. This leads to excellent results using a small number of refinement steps. In some sense, this also takes away one of the key advantages of DPMs over GANs, which is DPMs minimize a well-defined objective function. Nevertheless, the results are convincing, but not spectacular. I am not convinced that we should continue to report training FID on CIFAR-10. I would have like to see class-conditional ImageNet results. Also, it is not clear whether the proposed technique provides additional gains on top of SoTA GANs. Overall, I recommend acceptance as a spotlight.